# Autocrine INSL5 promotes tumor progression and glycolysis via activation of STAT5 signaling

Shi-Bing Li[1,†], Yan-Yan Liu[2,†], Li Yuan[1,†], Ming-Fang Ji[3,†], Ao Zhang[1], Hui-Yu Li[4], Lin-Quan Tang[1,5], Shuo-Gui Fang[1,6], Hua Zhang[7], Shan Xing[1], Man-Zhi Li[1], Qian Zhong[1] (iD), Shao-Jun Lin[8], Wan-Li Liu[1], Peng Huang[1], Yi-Xin Zeng[1], Yu-Ming Zheng[9,‡], Zhi-Qiang Ling[10,‡], Jian-Hua Sui[4,‡] & Mu-Sheng Zeng[1,*,‡] (iD)

## Abstract

Metabolic reprogramming plays important roles in development and progression of nasopharyngeal carcinoma (NPC), but the underlying mechanism has not been completely defined. In this work, we found INSL5 was elevated in NPC tumor tissue and the plasma of NPC patients. Plasma INSL5 could serve as a novel diagnostic marker for NPC, especially for serum VCA-IgA-negative patients. Moreover, higher plasma INSL5 level was associated with poor disease outcome. Functionally, INSL5 overexpression increased, whereas knockdown of its receptor GPCR142 or inhibition of INSL5 reduced cell proliferation, colony formation, and cell invasion *in vitro* and tumorigenicity *in vivo*. Mechanistically, INSL5 enhanced phosphorylation and nuclear translocation of STAT5 and promoted glycolytic gene expression, leading to induced glycolysis in cancer cells. Pharmaceutical inhibition of glycolysis by 2-DG or blockade of INSL5 by a neutralizing antibody reversed INSL5-induced proliferation and invasion, indicating that INSL5 can be a potential therapeutic target in NPC. In conclusion, INSL5 enhances NPC progression by regulating cancer cell metabolic reprogramming and is a potential diagnostic and prognostic marker as well as a therapeutic target for NPC.

**Keywords** diagnosis; glycolysis; INSL5; nasopharyngeal carcinoma; STAT5
**Subject Categories** Cancer; Metabolism; Molecular Biology of Disease

## Introduction

Cancer cells require abundant metabolic intermediates and energy to fuel cell growth and division (Cairns *et al*, 2011). To meet these elevated requirements, cancer cells often take up large amounts of glucose and generally limit their energy metabolism to glycolysis for ATP generation despite the presence of oxygen, which is termed aerobic glycolysis (also called the Warburg effect; Hsu & Sabatini, 2008; Vander Heiden *et al*, 2009; Schulze & Harris, 2012). This metabolic shift toward aerobic glycolysis enables cancer cells to convert glucose more efficiently into macromolecules, which are needed for rapid cell growth. On the one hand, cancer cells enhance glycolytic flux by upregulating the expression of key glycolytic genes, including the glucose transporters Glut1 and Glut3 (Ha *et al*, 2012; Kuang *et al*, 2017), hexokinase (HK2) (Patra *et al*, 2013; Yang *et al*, 2018), lactate dehydrogenase (LDH; Faubert *et al*, 2017), and phosphofructokinase-1 (PFK1; Krall & Christofk, 2017; Peeters *et al*, 2017). On the other hand, cancer cells can increase the enzymatic activity of some key enzymes to promote the production of metabolic intermediates(Shaul *et al*, 2014; Haas *et al*, 2016). Oncogenic viruses have been reported to alter the cellular metabolic pathway in cancer cells (Mesri *et al*, 2014; Levy *et al*, 2016; Zhu *et al*, 2016). Therefore, identification of the metabolic linker between virus infection and cancer cells could present new opportunities to target oncogenic virus-related cancer.

Nasopharyngeal carcinoma (NPC) is a malignant epithelial tumor with a unique geographic distribution, as it is mainly found in Southern China and South-East Asia (Chan *et al*, 2002). NPC is etio-

1 State Key Laboratory of Oncology in South China, Guangdong Key Laboratory of Nasopharyngeal Carcinoma Diagnosis and Therapy, Collaborative Innovation Center for Cancer Medicine, Sun Yat-sen University Cancer Center, Guangzhou, China
2 Department of Nephrology, Division of Internal Medicine, Tongji Hospital, Tongji Medical College, Huazhong University of Science and Technology, Wuhan, China
3 Cancer Research Institute of Zhongshan City, Zhongshan, China
4 National Institute of Biological Sciences, Beijing, China
5 Department of Nasopharyngeal Carcinoma, Sun Yat-sen University Cancer Center, Guangzhou, China
6 Department of Radiation Oncology, Sun Yat-sen University Cancer Center, Guangzhou, China
7 School of Medicine, Sun Yat-sen University, Guangzhou, China
8 Department of Radiation Oncology, Fujian Provincial Cancer Hospital, Fuzhou, China
9 Department of Clinical Laboratory, Wuzhou Red Cross Hospital, Wuzhou, China
10 Zhejiang Cancer Hospital, Hangzhou, China
*Corresponding author. Tel: +86 208734 3191; E-mail: zengmsh@mail.sysu.edu.cn or zengmsh@sysucc.org.cn
†These authors contributed equally to this work
‡These authors contributed equally to this work as senior authors

logically associated with Epstein–Barr virus (EBV) infection (Raab-Traub, 2002; Young & Rickinson, 2004; Tsao et al, 2015). In NPC, EBV mainly expresses latent genes, including latent membrane protein 1 (LMP1), Epstein–Barr nuclear antigen 1 (EBNA1), latent membrane protein 2A (LMP2A), and EBV-encoded small non-polyadenylated RNAs (EBERs; Pathmanathan et al, 1995; Nanbo & Takada, 2002; Arvey et al, 2012; Tsang et al, 2014). In addition to latent gene expression, EBV lytic gene expression has also been demonstrated in NPC tumors in recent studies by transcriptomic sequence analysis (Hu et al, 2016). Previous studies showed that EBV-encoded latent genes, such as LMP1, could reprogram the glucose metabolism of NPC cells, illustrating the significance of metabolic reprogramming in the development of NPC (Xiao et al, 2014; Lo et al, 2015). However, the potential major molecular targets involved in modulating NPC metabolism remain to be identified.

Insulin-like peptide 5 (INSL5) is a member of the relaxin/insulin superfamily and presents a tertiary structure similar to that of other insulin family members (Conklin et al, 1999). Mature INSL5 consists of two chains (an A-chain and a B-chain) that are linked by two disulfide bonds with a third intramolecular disulfide within the A-chain (Akhter Hossain et al, 2008; Haugaard-Jonsson et al, 2009; Luo et al, 2010; Belgi et al, 2011). INSL5 functions by binding to its receptor, G protein-coupled receptor 142 (GPCR142/RXFP4). The positively charged B-chain residues (B13Arg and B23Arg) of INSL5 and the negatively charged extracellular residues (Glu100, Asp104, and Glu182) of GPCR142 are important for INSL5-GPCR142 binding and activation (Wang et al, 2014). The primary functions of INSL5 have been suggested to affect glucose metabolism and reproductive physiology in a mouse model, as $Insl5^{-/-}$ mice exhibit impaired male and female fertility due to a marked reduction in sperm motility and irregular length of the estrous cycle, respectively (Burnicka-Turek et al, 2012; Grosse et al, 2014; Lee et al, 2016). In mice experiencing energy deprivation, Isnl5 contributes to appetite promotion and hepatic glucose production (Grosse et al, 2014). A recent report suggested an important role for Isnl5 in the regulation of insulin secretion and β-cell homeostasis (Luo et al, 2015). Other reports identified that colonic Isnl5 expression is reduced by the gut microbiota and energy availability (Lee et al, 2016). In human diseases, INSL5 has been identified recently in colonic tissue and neuroendocrine tumors(Thanasupawat et al, 2013), but its specific functions in tumors remain to be elucidated.

In this study, we identified INSL5 was upregulated after EBV infection in nasopharyngeal epithelial cells. INSL5 was overexpressed in NPC and correlated with poor prognosis of the patients. We found that INSL5 could promote glycolytic gene expression and NPC progression by activating the GPCR142/STAT5 axis. Importantly, inhibition of INSL5 function or its downstream factors by various methods reversed INSL5-induced cancer progression in vitro and in vivo, suggesting INSL5 is a potential therapeutic target for NPC.

# Results

## INSL5 is highly expressed in NPC tumors

Virus infection can skew metabolism toward glycolysis over oxidative phosphorylation (OXPHOS) in a manner similar to the Warburg effect in cancer (Claus & Liebert, 2014; Zhang et al, 2017). To examine whether tumorigenic virus-induced metabolic alterations can contribute to tumor progression, we performed a gene microarray to analyze the metabolism-associated genes differentially expressed between EBV-infected and uninfected cells in a previously established high-efficiency EBV infection model. INSL5 was among the metabolism-associated genes upregulated after EBV infection (Fig EV1A). The upregulation of INSL5 by EBV was verified in two additional EBV-infected NPC cell lines (Fig EV1B). Although INSL5 has been reported to play an important role in glucose homeostasis in mice, there is little knowledge about its roles in virus infection and cancer progression. Thus, we selected INSL5 for further study.

We then first determine INSL5 protein levels in NPC cell lines, tumor, and plasma samples of NPC patients. We found that INSL5 expression was higher in most of the NPC cell lines than in the three immortalized nasopharyngeal epithelial cells (NPECs) by Western blotting (Fig 1A). By qRT–PCR analysis, INSL5 mRNA levels were significantly increased in a cohort of 62 NPC biopsy samples than the cohort of 48 normal control samples (NPN; Fig EV1C). In consistent, we found INSL5 expression was mainly detected in cancer cells rather than in stromal cells or normal epithelial cells in 18 NPC tumor samples by immunohistochemistry (Fig EV1D). Collectively, these data indicated that INSL5 was highly expressed in NPC tumor cell lines and tumor tissues.

## Plasma INSL5 is a diagnostic biomarker for NPC patients

As INSL5 is a secreted protein, we established an ELISA kit to detect secreted INSL5 in cell culture supernatants and the plasma of NPC patients. We found that the NPC cell lines secreted higher INSL5 levels in the culture supernatants than the immortalized NPECs did (Fig 1B). Importantly, NPC patients (n = 159) presented higher plasma INSL5 concentration (median, 2.50 ng/ml, quartile, 1.47–3.52) than the VCA-IgA-positive (n = 54, median, 0.66 ng/ml; quartile, 0.31–1.26) or VCA-negative (n = 44, median, 0.49 ng/ml; quartile, 0.33–0.91) control groups (Fig 1C) in the training cohort. Compared with the VCA-IgA-negative control group, the VCA-IgA-positive control group presented a significantly elevated INSL5 level (Fig 1C). The plasma INSL5 level could have significant diagnostic potential in NPC (AUC = 0.881, 95% CI: 0.835–0.926) (Fig 1D). Additionally, similar results were confirmed in two validation cohorts, validation cohort 1 (group EBV (−): median 0.47 ng/ml, quartile 0.36–0.70; group EBV (+): median 2.23 ng/ml, quartile 2.10–2.83; group NPC: median 4.13 ng/ml, quartile 3.13–5.06; Fig 1E) and validation cohort 2 (group EBV (−): median 0.27 ng/ml, quartile 0.18–0.58; group EBV (+): median 0.55 ng/ml, quartile 0.35–0.92; group NPC: median 1.64 ng/ml, quartile 1.13–2.31) (Fig 1F). Applying the cut-off value (1.12 ng/ml) of the training cohort to the validation cohorts for NPC diagnosis yielded sensitivities of 97.1% and 76.1% and specificities of 73.4% and 87.0%, respectively (Fig 1G). Combining all the samples, we found that this cut-off value could differentiate NPC patients from normal controls with a sensitivity of 89.6% and a specificity of 77.7% (Fig 1G), and the INSL5 level demonstrated impressive sensitivity and specificity for NPC diagnosis (AUC 0.914, 95% CI: 0.897–0.931) (Fig 1H), which is comparable to VCA-IgA for NPC diagnosis.

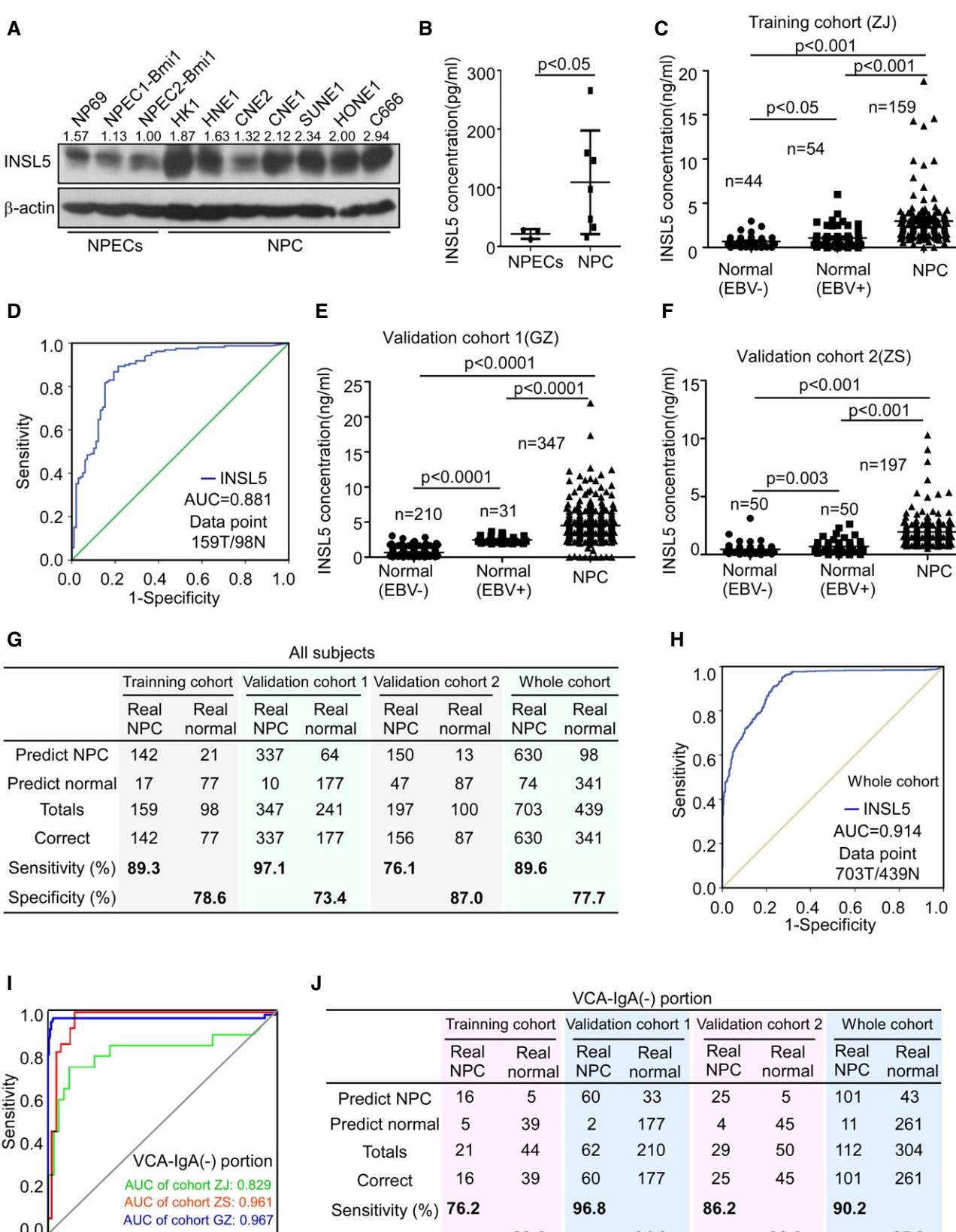

Figure 1.

◀

**Figure 1.  INSL5 is elevated in NPC tissue and plasma and is a potential diagnostic biomarker.**

A   Western blotting showing cellular INSL5 in the immortalized NPEC and NPC cell lines after inhibition of protein secretion by BFA, and β-actin was used as a loading control.

B   ELISA assay showing secreted INSL5 in the supernatants of NPECs and NPC cell lines. For NPECs, $n = 3$; for NPC, $n = 7$.

C   The concentration of plasma INSL5 in healthy controls (normal EBV (−)), non-tumor individuals with VCA-IgA positive (normal EBV (+)), and NPC patients, in training cohort.

D   ROC of the diagnostic prediction model with plasma INSL5 level in the training cohort.

E   The concentration of plasma INSL5 in healthy controls (normal EBV (−)), non-tumor individuals with VCA-IgA positive (normal EBV (+)), and NPC patients, in validation cohort 1(GZ).

F   The concentration of plasma INSL5 in healthy controls (normal EBV (−)), non-tumor individuals with VCA-IgA positive (normal EBV (+)), and NPC patients, in validation cohort 2(ZS).

G   Confusion table of binary results of the diagnostic prediction model in the training cohort, validation cohorts, and the whole cohort.

H   ROC of the diagnostic prediction model with plasma INSL5 level in the whole cohort.

I   ROC of the diagnostic prediction model with plasma INSL5 level in individuals with VCA-IgA-negative plasma in all three cohorts.

J   Confusion table of binary results of the diagnostic prediction model in VCA-IgA-negative individuals in the training cohort, validation cohorts, and the whole cohort.

Data information: In (B), data are presented as mean ± SD, in (C, E and F), data are presented as mean ± SEM, and *P*-values were determined by unpaired *t*-test.
*$P < 0.05$, **$P < 0.01$, ***$P < 0.001$, ns, no significance. Exact *P*-values are specified in Appendix Table S4.
Source data are available online for this figure.

## Plasma INSL5 is a diagnostic biomarker to distinguish EBV seronegative NPC patients from normal controls

Although VCA-IgA is the widely used marker for NPC diagnosis, there are still 4–24% of patients with VCA-IgA undetectable. We found that INSL5 had especially high diagnostic efficiency (training cohort: AUC = 0.829, 95% CI 0.695–0.963; validation cohort 1: AUC = 0.961, 95% CI 0.919–1.00; and validation cohort 2: AUC = 0.967, 95% CI 0.924–1.00) in individuals with VCA-IgA-negative plasma in all three cohorts (Fig 1I). Applying the same cut-off value (1.12 ng/ml) to all VCA-IgA-negative individuals yielded a sensitivity of 90.2% and a specificity of 85.9% for NPC diagnosis (Fig 1J). In summary, plasma INSL5 could be a novel diagnostic marker for NPC patients, especially to assist for the diagnosis of VCA-IgA-negative patients.

## Plasma INSL5 is a prognostic marker for NPC patients

To determine whether plasma INSL5 could be a prognostic marker for NPC patients, we preformed receiver operating characteristic (ROC) curve analysis to identify the optimum cut-off value (3.73 ng/ml) for plasma INSL5 level for prognosis analysis and used the Kaplan–Meier test to analyze the correlation between the concentration of INSL5 and patient survival in GZ cohort which is the only one cohort with complete clinical outcome information. The characteristics of the three cohorts' patients with NPC are listed in Appendix Table S1. As shown, the patients with high INSL5 concentrations had shorter overall survival (OS) ($P = 0.005$; Fig 2A) and disease-free survival (DFS) ($P = 0.028$) than those with low INSL5 concentrations (Fig 2D). In addition, as the traditional biomarker for progression prediction (Li *et al*, 2017), a higher EBV DNA copy number also indicated shorter OS ($P = 0.038$; Fig 2B) and DFS ($P = 0.004$) than a lower copy number (Fig 2E). Furthermore, when we combined the INSL5 level and EBV DNA copy number, we found that the patients with high INSL5 levels and EBV DNA copy numbers had significantly shorter OS ($P = 0.003$) (Fig 2C) and DFS ($P = 0.002$) than the other patients (Fig 2F). Additionally, the profiles from TCGA database also showed that high INSL5 expression was correlated with poor overall survival in multiple tumors, especially in glioma, kidney renal clear cell carcinoma, sarcoma, uterine carcinosarcoma, and uveal melanoma patients (Fig EV1E–I).

Next, we performed univariate and multivariate analyses using the Cox proportional hazards model to determine whether the INSL5 plasma level could serve as an independent prognostic predictor. A series of factors, including patient age, gender, body mass index (BMI), T classification, N classification, EBV DNA copy number, LDH concentration, hs-CRP concentration, VCA-IgA titer, EA-IgA titer, progression status, and INSL5 concentration, were assessed in the univariate Cox regression analysis to test their association with the OS and DFS of NPC patients. As shown in Appendix Table S2, INSL5 plasma concentration was indeed a prognostic factor for OS and DFS. Additionally, EBV DNA copy number, BMI, and hs-CRP were prognostic factors for OS, and EBV DNA, LDH, and VCA-IgA were prognostic factors for DFS. The variables most significantly associated with OS or DFS in the univariate analysis were further analyzed by multivariate analysis. The multivariate analysis model revealed that the predominant independent predictor of OS and PFS was INSL5 concentration, as shown in Appendix Table S2. A nomogram is an efficient model to assess the efficiency of several independent factors for prognostic prediction based on the results of multivariable Cox regression analysis. First, we built a nomogram to predict OS using the variables INSL5 level, EBV DNA copy number, BMI, CRP expression, T stage, N stage, and age (Fig 2G). In the DFS prediction analysis, we identified that INSL5 expression, age, LDH expression, EBV DNA copy number, and VCA-IgA status could be the variables in the nomogram (Fig 2H). The calibration plot also showed good agreement between the predictions by the nomogram for OS or DFS and the actual observations (Fig EV1J and K). Collectively, these data suggested that INSL5 may contribute to risk stratification and prognosis prediction in patients with NPC. Patients with higher INSL5 levels showed poorer survival.

## INSL5 promotes cell proliferation and invasion through an autocrine model of action

To determine the effects of INSL5, stable NPC cell lines (CNE1, CNE2, HK1) and a normal cell line (NP69) were established via retroviral infection (Fig 3A upper panel and Fig EV2A). Cytoplasmic localization of overexpressed INSL5 was observed after treatment with the protein secretion inhibitor BFA (Fig 3A, lower panel). INSL5 overexpression significantly accelerated the proliferation of the NPC cells (Fig 3B, upper panel), which was almost completely

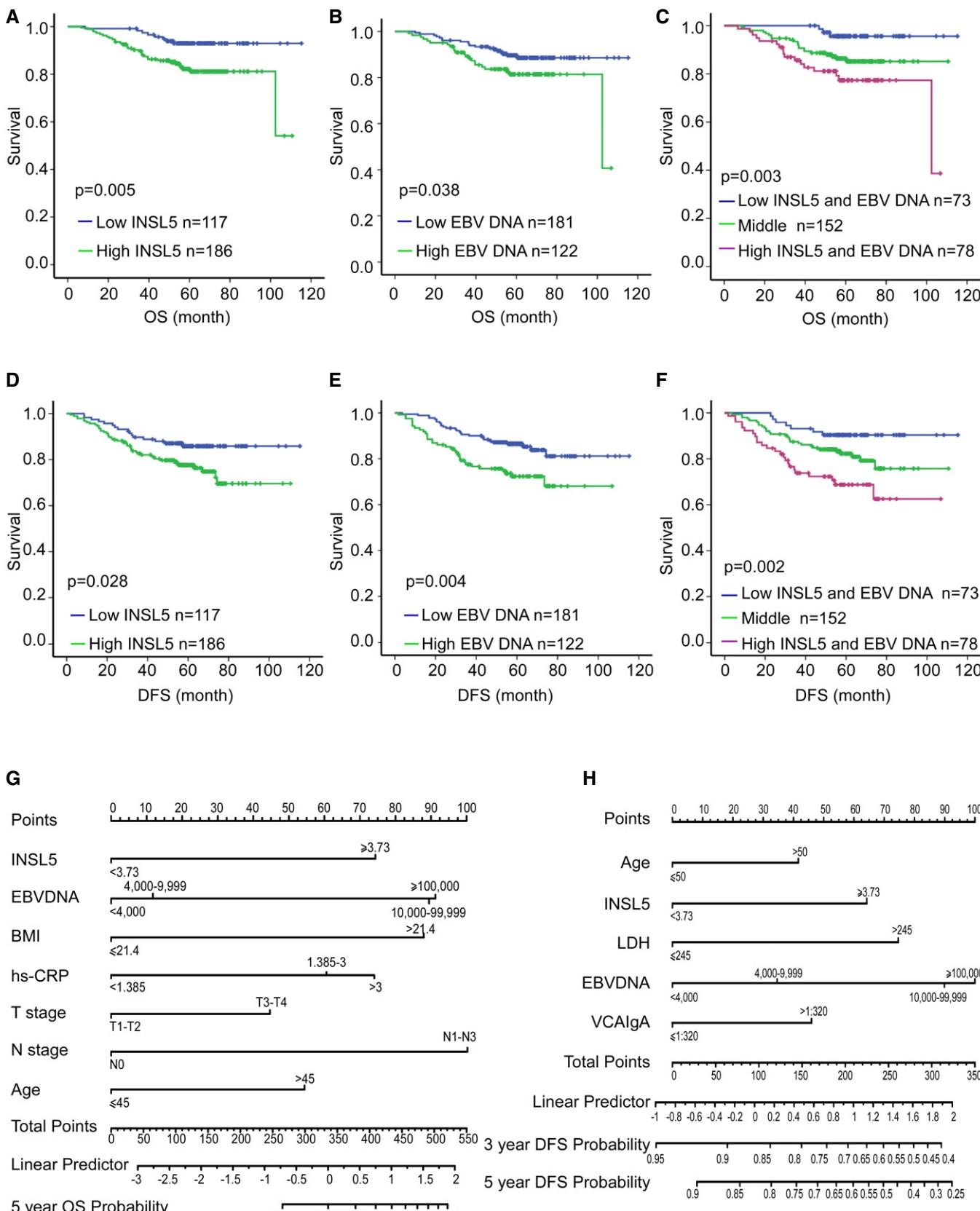

**Figure 2.**

Figure 2. Higher plasma INSL5 is associated with poor prognosis of NPC patients.

A–C Kaplan–Meier curves showing the overall survival (OS) curves of NPC patients with low or high INSL5 concentration (A), low (≤ 4,000 copy/ml), or high (> 4,000 copy/ml) EBV DNA copy number (B), and the combination of INSL5 concentration with EBV DNA copy number (C).
D–F Kaplan–Meier curves showing the disease-free survival (DFS) curves of NPC patients with low or high INSL5 concentration (D), low or high EBV DNA copy number (E), and the combination of INSL5 concentration with EBV DNA copy number (F).
G Nomogram, including plasma INSL5 level, plasma EBV DNA, body mass index (BMI), pretreatment hs-CRP, T stage, N stage, and age for 5-year overall survival (OS) in patients with nasopharyngeal carcinoma.
H Nomogram, including age, plasma INSL5 level, serum lactate dehydrogenase level, plasma EBV DNA, and VCA-IgA titer, for three- and 5-year disease-free survival (DFS) in patients with nasopharyngeal carcinoma.

Data information: In (A-F), $P$-values were determined by log-rank test, $n = 303$.

reversed by siRNA silencing of the INSL5 receptor GPCR142 (Fig 3B, lower panel) and the knockdown efficiency of GPCR142 was detected by qRT–PCR (Fig EV2B). In addition, INSL5 overexpression also promoted cell proliferation in normal nasopharyngeal epithelia cell (Fig EV2C). Similarly, colony formation, BrdU incorporation, migration, and invasion were all promoted by the overexpression of INSL5, and all these effects were reversed by siRNA knockdown of GPCR142 expression (Figs 3C–H and EV2D–G).

To further explore the function of endogenous INSL5 in NPC, we utilized a specific siRNA to knockdown endogenous INSL5 expression in two different EBV-positive cell lines (HNE1-EBV and CNE2-EBV). Both siRNAs reduced the INSL5 mRNA and protein levels by 50–70% in the different cell lines (Fig EV2H). Consistent with the observations in the INSL5-overexpressing cells, knockdown of INSL5 expression by siRNA significantly decreased the proliferation and colony formation ability as well as the migration and invasion of NPC cells (Fig EV2I and J). To examine the effect of INSL5 on the tumorigenicity of NPC cells *in vivo*, INSL5-overexpressing cells (HK1 and CNE2 cell lines) and control cells were subcutaneously injected into nude mice. As shown, the growth rate, tumor volume (Figs 3I and EV2K), and tumor weight (Figs 3J and EV2L) were dramatically enhanced in the INSL5 overexpression group compared with the control group. Collectively, the results show INSL5 promotes NPC progression by enhancing cancer cell proliferation and invasion abilities through an autocrine model of action, which depends on the expression of the INSL5 receptor GPCR142 on cancer cells.

## INSL5 induces metabolic reprogramming in NPC cells manifested by active glycolysis

To explore the mechanism by which INSL5 promotes NPC progression, we performed an mRNA microarray to investigate the gene expression profile affected by ectopic INSL5 expression. Hierarchical clustering revealed all the upregulated and downregulated genes in the CNE2 cell line that stably overexpressed INSL5 (Fig 4A). Gene Set Enrichment Analysis (GSEA) showed that INSL5 overexpression primarily affected the signaling pathway gene sets, including those for glycolysis, the PI3K-Akt signaling pathway, and the cell cycle, resulting in remarkable changes in the biological processes of cell proliferation and glucose metabolism (Fig 4B). Among the glucose metabolic signals, OXPHOS had a negative correlation with INSL5 overexpression identified by GSEA (Fig 4C). Therefore, we focused on the altered metabolic genes that were upregulated by INSL5. First, we verified the microarray results by qPCR and found that INSL5-mediated glycolytic alterations by regulating key enzymes at the transcriptional level (Fig EV3A). Consistently, these enzymes were also upregulated at the protein level in the INSL5-overexpressing cells under conditions of nutrient stress (Fig 4D). In contrast, silencing endogenous INSL5 expression decreased the expression of these enzymes (Fig 4D). Additionally, knockdown of its receptor GPCR142 could reverse INSL5-induced glycolytic gene expression (Fig EV3B). We then performed a liquid chromatography-mass spectrometry (LC-MS)-based metabolomic analysis to detect the intermediates of glucose metabolism. As shown, INSL5 overexpression mainly increased glycolytic metabolite levels and also enhanced the intermediates involved in pentose phosphate pathway (PPP), nucleotide synthesis, and amino acid production, while decreased the levels of intermediates in the tricarboxylic acid cycle (TCA cycle; Figs 4E and EV3C–E). Together, we concluded that INSL5 could enhance key glycolytic gene expression with a concomitant increase in metabolic intermediates (Fig EV3F), indicating that INSL5 mediates glycolytic reprogramming in cancer cells. To verify the role of INSL5 in metabolic reprogramming, the extracellular acidification rate (ECAR) and oxygen consumption rate (OCR) were measured using a Seahorse XF96 Extracellular Flux analyzer. We demonstrated that INSL5 overexpression significantly increased glycolysis

Figure 3. INSL5 promotes the progression of NPC via accelerating cell proliferation and invasion depending on GPCR142.

A Exogenous expression of INSL5 in NPC cells. Representative immunoblotting (upper panel) and immunofluorescent staining (lower panel) showed stable exogenous expression of INSL5 in both CNE1, CNE2, and HK1 NPC cell lines. Scale bars represent 20 μm.
B MTT assay of vector control or INSL5 overexpressing CNE1, CNE2, and HK1 NPC cell lines (upper panel) either transfected with control siRNA (NC) or GPCR142 siRNA (#1 and #2) (lower panel). $n = 4$ biological replicates for CNE1 and CNE2 cell line, $n = 6$ biological replicates for HK1.
C–H Colony formation (C and D), Brdu incorporation (E and F), and migration assays (G and H) of vector control or INSL5 overexpressing CNE1, CNE2, and HK1 NPC cell lines either transfected with control siRNA (NC) or GPCR142 siRNA (#1 and #2). Representative images are shown in (C), (E), and (G) for colony formation, Brdu incorporation, and migration assays, respectively. Number of colonies, the percentage of Brdu positive cells, and migrated cells per field of view were plotted in (D, F, and H), respectively. The results are from three different experiments. Scale bars represent 20 μm in (E) and 100 μm in (G).
I, J Xenograft tumor growth of INSL5 overexpression NPC HK1 stable cell lines in nude mice. Tumor size (I) and tumor weight (J) of two groups. $n = 11$ mice per group.

Data information: In (B, I, and J), data are presented as mean ± SD, in (D, F, and H), data are presented as mean ± SEM, from three different experiments, and $P$-values were determined by unpaired $t$-test. *$P < 0.05$, **$P < 0.01$, ***$P < 0.001$, ns, no significance. Exact $P$-values are specified in Appendix Table S4.
Source data are available online for this figure.

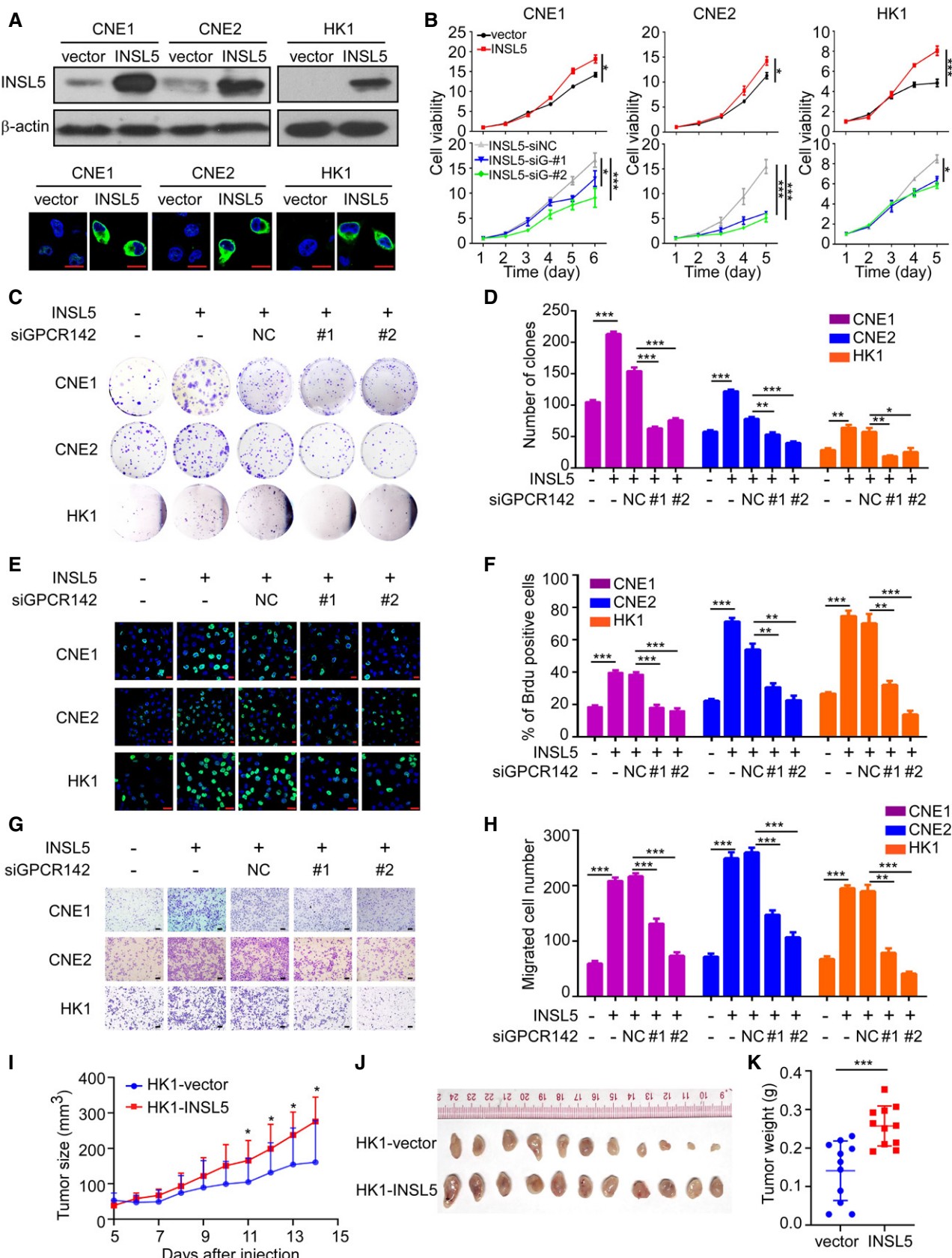

Figure 3.

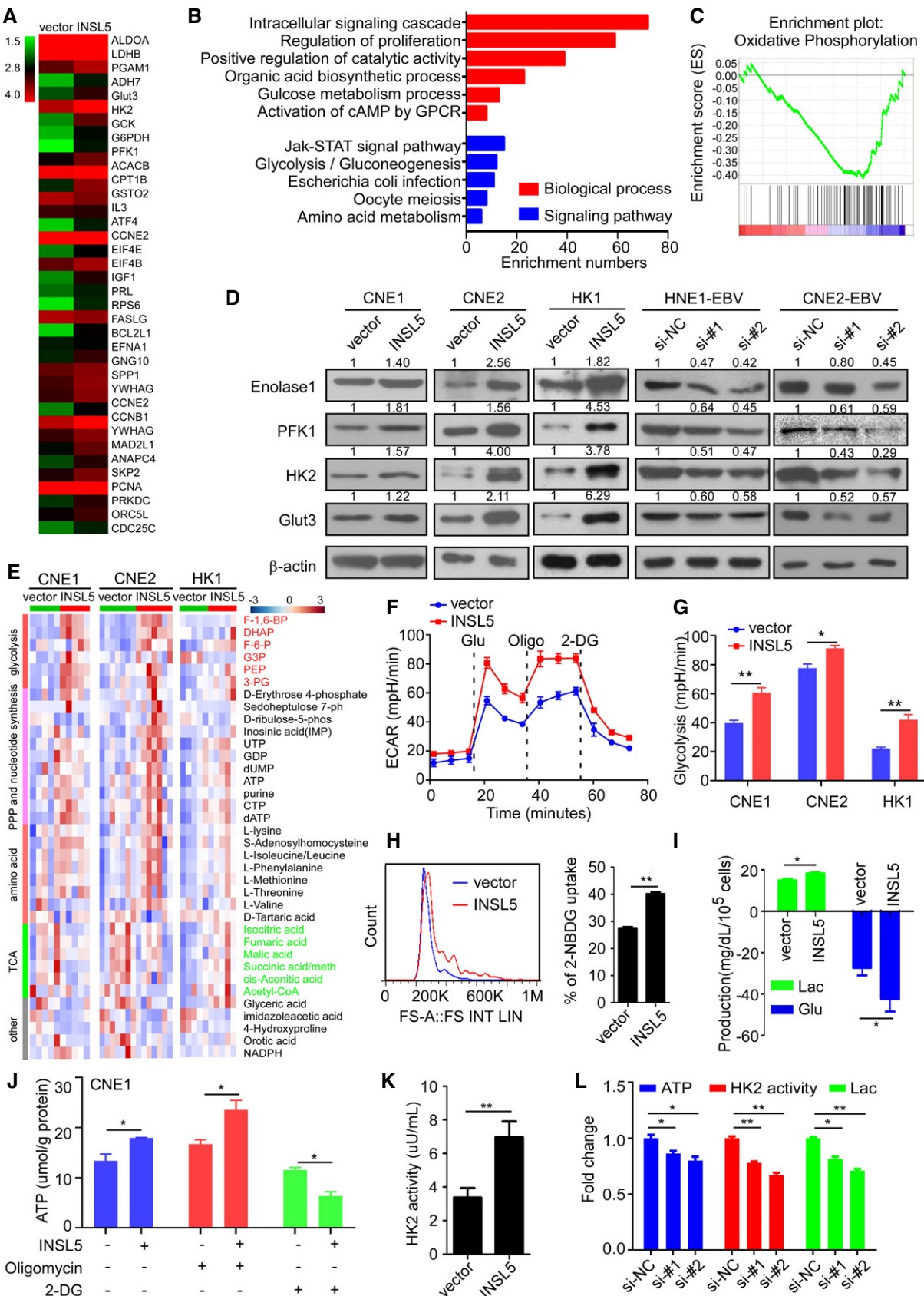

Figure 4.

**Figure 4. INSL5 induces glucose metabolism to aerobic glycolysis reprogramming in NPC cells.**

A Gene expression profiles from CNE2 cells overexpressed with INSL5 or the vector control. The genes are shaded with green, black, or red in the heat map to indicate low, intermediate, or high expression, respectively.

B Functional annotation clustering of genes regulated by INSL5 in CNE2 cells is shown. Significantly enriched groups nominated by the gene ontology term are ranked based on the group enrichment scores. Red: biological process. Blue: signaling pathway.

C The signal pathway of the subset of genes involved in metabolism was assessed by Gene Set Enrichment Analysis.

D Analysis of glycolytic gene expression by immunoblot in INSL5 overexpressed or knockdown cell lines.

E Metabolites were purified from five (CNE1 cell line and HK1 cell line) or seven (CNE2 cell line) independent samples of two vector and INSL5 overexpressed cell lines and analyzed by liquid chromatography-mass spectrometry (LC-MS). Red: glycolytic intermediate metabolites. Green: tricarboxylic acid cycle (TCA cycle) intermediate metabolites. PPP: pentose phosphate pathway. The color scale bar represents Z-score of the metabolites' level.

F The extracellular acidification rate (ECAR) was measured in cells with or without INSL5 overexpression using a Seahorse XF96 Extracellular Flux analyzer. Glu, glucose; Oligo, oligomycin.

G Statistical analysis of the effects of INSL5 overexpression on glycolytic activity.

H CNE1-vector and CNE1-INSL5 cells were grown in the presence of the fluorescent glucose analog 2-NBDG (Invitrogen) for 15 min, and glucose uptake was then quantified using flow cytometry (FACS) (left panel). Statistical analysis of the glucose uptake was showed in the right panel.

I–K Glucose consumption and lactate production (I), basic ATP level or the ATP level of cell which was treated by 1 μM oligomycin or 50 μM 2-DG for 24 h (J), HK2 enzyme activity (K) in CNE1-vector and CNE1-INSL5 cells.

L ATP concentration, HK2 enzyme activity, and lactate production in INSL5 wide-type or knockdown HNE1-EBV cells.

Data information: In (F, G, and L), data are presented as mean ± SEM, in (H–K), data are presented as mean ± SD, from three different experiments, and *P*-values were determined by unpaired *t*-test. *$P < 0.05$, **$P < 0.01$, ***$P < 0.001$, ns, no significance. Exact *P*-values are specified in Appendix Table S4.

Source data are available online for this figure.

(Figs 4F and G, and EV3G) and impaired OXPHOS (Fig EV3H), which is consistent with metabolomic results. To determine whether INSL5 influences NPC cell "appetite" in a cell-autonomous fashion, we used flow cytometry to measure glucose uptake in INSL5-overexpressing cell lines using a fluorescent glucose analog (2-NBDG) that is incorporated into cells and allows quantification of glucose uptake. Notably, INSL5-overexpressing cells display a striking increase in glucose uptake (Figs 4H and EV3I). The increased glucose uptake and altered enzymes prompted us to test how glucose is utilized in INSL5-overexpressing or INSL5-deficient cells. We first measured lactate production to determine whether glycolysis was enhanced. Indeed, INSL5-overexpressing cells displayed significantly higher glucose consumption and higher levels of lactate production than control cells (Figs 4I and EV3L), indicating that cells switch to glycolysis to sustain ATP production under conditions of nutrient stress. Next, we tested the effect of INSL5 on HK2 activity and ATP production. We identified that INSL5 prominently increased the production of ATP (Figs 4J and EV3K) and the activity of HK2 (Figs 4K and EV3J). When we treated the cells with OXPHOS inhibitor (oligomycin) or glycolysis inhibitor (2-DG), we found that oligomycin treatment slightly increased the ATP level, while 2-DG treatment greatly decreased the ATP level, INSL5 overexpressing cells especially displayed a greater extent reduction (Figs 4J and EV3K), which indicated that INSL5 overexpression indeed promoted glucose metabolism shift from OXPHOS to glycolysis. What's more, silencing its receptor GPCR142 will obviously decrease glycolysis induced by INSL5 (Fig EV3M), which indicated that INSL5 reprogrammed glucose metabolism dependent on the identified receptor, GPCR142. In addition, the knockdown of endogenous INSL5 decreased the glycolysis phenomenon (Figs 4L and EV3N). Overall, these results indicated that the presence of INSL5 enhanced glycolysis in cancer cells.

## Activation of STAT5 by INSL5 facilitates aerobic glycolysis and proliferation

Based on the upregulated metabolic enzyme expression and the enhancement of glycolysis by INSL5, we wanted to know how

INSL5 affects the expression of key enzymes at the transcriptional level. By analyzing promoter sequences, we found that key enzymes, including HK2, Glut3, and PFK1, contained the same Ebox STAT5 binding site (Fig 5A). Therefore, we hypothesized that INSL5 may activate STAT5 to promote enzyme expression. Previous evidence suggested that active STAT5 exhibited a phosphorylated form when cells were stimulated with cytokines and growth hormones (Bowman *et al*, 2000; Mziaut *et al*, 2006; Miklossy *et al*, 2013). The active phosphorylated STAT5 proteins form dimers in the cytosol and then move into the nucleus, where they bind to chromatin and activate targeted gene transcription (Cotarla *et al*, 2004; Yu & Jove, 2004). We found that the INSL5 peptide could promote the phosphorylation of STAT5, which was diminished by an anti-INSL5 neutralizing antibody (Fig 5B). These findings were corroborated by immunofluorescence microscopy, which demonstrated that INSL5 could enhance the nuclear localization of STAT5 (Fig 5C). Additionally, the analysis of separate cytosolic and nuclear fractions also showed greater nuclear accumulation of phosphorylated STAT5 after INSL5 stimulation (Fig 5D). All of these data suggested that INSL5 indeed promoted the activation of STAT5. It is reported that INSL5 could active Akt and ERK1/2 signal (Ang *et al*, 2017, 2018), which may be the upstream of STAT5. Then, we detected the activation of ERK1/2, Akt, and JAK1, found that INSL5 could promote the activation of ERK1/2, Akt, and JAK1. Meanwhile, we detected the phosphorylation of STAT5 in cells treated with different kinase inhibitors and found that JAK1 (Ruxolitinib) and ERK1/2/2 (U0126) inhibitors, but not Akt inhibitor (MK2206), could reverse STAT5 activation. STAT3 activation is not induced by INSL5 overexpression (Fig 5E). Taken together, those results indicated that JAK and ERK1/2 kinase contributed to STAT5 activation induced by INSL5/RXFP4. We next investigated whether STAT5 enhanced the expression of HK2, Glut3, and PFK1 under stimulation with INSL5. A STAT5 ChIP assay was performed with the INSL5-overexpressing CNE1 cell line. The enrichment of the STAT5 binding site in each gene was then determined by qPCR, which showed that INSL5 enhanced the binding of STAT5 to the promoters of HK1, Glut3, and PFK (Fig 5F). Consistently, the luciferase assay also demonstrated that STAT5 could induce the promoter activity of

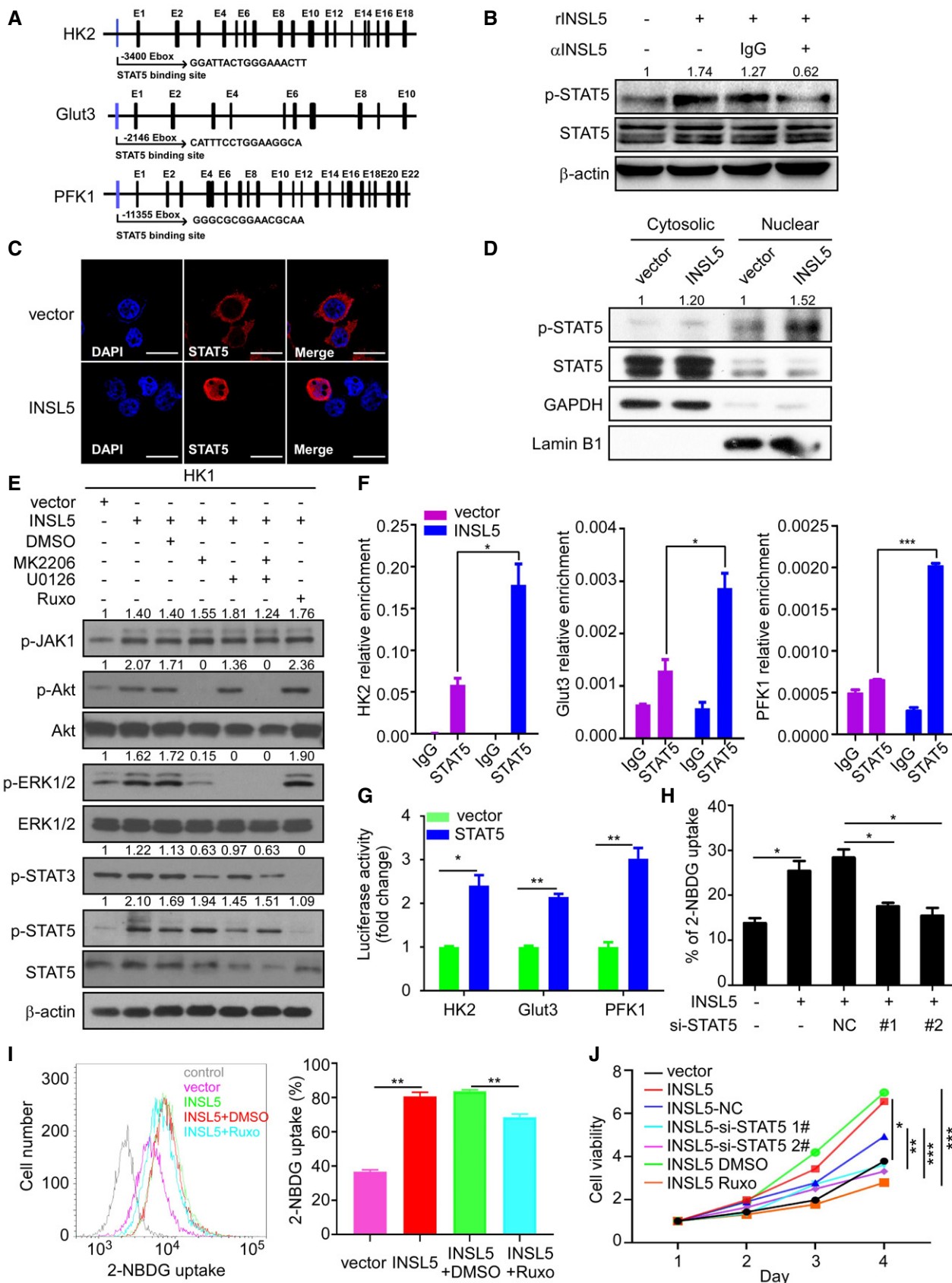

Figure 5.

**Figure 5. Activation of STAT5 by INSL5 facilitates aerobic glycolysis and proliferation.**

A  Schematic diagram of the mRNA structures of the glycolytic genes and the indicated STAT5 binding site.
B  Phosphorylation of STAT5 under INSL5 stimulation. CNE1 cells were starved in FBS-free medium for 24 h and then treated with INSL5 peptide (50 ng/ml) in the presence of an anti-INSL5 antibody at 100 μg/ml, or an isotype-matched IgG control (IgG) for 30 min.
C  Localization of STAT5. A confocal analysis of STAT5 staining was performed using CNE1 cells transfected vector or INSL5. Scale bars represent 20 μm.
D  Cytoplasmic and nuclear proteins from vector or INSL5 overexpressing cells were separated and subjected to immunoblot as indicated. The nuclear marker Lamin B1 and the cytoplasmic marker GAPDH were used to demonstrate the purity of fractions.
E  INSL5 overexpression activated AKT, ERK, JAK1, and STAT5 phosphorylation, not STAT3 in HK1 cells. INSL5 overexpressing HK1 cells were treated with MK2206 (50 μM), U0126(20 μM), MK2206 + U0126, or Ruxolitinib (5 μM) for 24 h. Western blotting was performed to evaluate the effects of those inhibitors on STAT5 and STAT3 phosphorylation level.
F  ChIP-qPCR analyzed the occupancy of potential E-boxes by STAT5 in the genes of HK2, Glut3 and PFK1 in INSL5-overexpressing CNE1 cells using IgG or anti-STAT5 antibodies.
G  Luciferase activity of different glycolytic gene promoter reporters in 293T cells transfected with STAT5 or empty vector.
H  Glucose uptake assessed by 2-NBDG incubation in INSL5 overexpressed CNE1 cells with STAT5 knockdown.
I  Glucose uptake was assessed by 2-BNDG incubation in INSL5 overexpressed HK1 cells treated with JAK kinase inhibitor (Ruxolitinib).
J  MTT assay of vector control or INSL5 overexpressing HK1 cells either treated with STAT5 siRNAs or JAK kinase inhibitor.

Data information: In (F–H and J), data are presented as mean ± SEM, in (I), data are presented as mean ± SD, from three different experiments, and *P*-values were determined by unpaired *t*-test. *$P < 0.05$, **$P < 0.01$, ***$P < 0.001$, ns, no significance. Exact *P*-values are specified in Appendix Table S4.
Source data are available online for this figure.

these metabolic enzymes (Fig 5G). The STAT5 pathway has been involved in tumorigenesis in several types of cancer (Miklossy *et al*, 2013; Able *et al*, 2017). Previous studies show that STAT5 can mediate oncogenic signals and regulate cell cycle progression, proliferation, and promote cancer cell survival. Therefore, we further investigated the effects of INSL5 on cell cycle progression and apoptosis. INSL5 overexpression in CNE1, CNE2, and HK1 cells strikingly promoted cell cycle progression (Fig EV4A and B). We also detected key cell cycle proteins and observed that INSL5 overexpression markedly increased cyclin E and cyclin D expression and decreased p27 expression, but only slightly increased cyclin B expression (Fig EV4C). What's more, STAT5 is known to activate anti-apoptotic genes, like c-myc, BCL2, and BCL-xL. We detected those genes in INSL5 overexpressing and control cells, and found that INSL5 overexpression only increased c-myc expression, but not BCL2 and BCL-xL (Fig EV4D). We also detected cell apoptosis under conventional chemotherapy (5-FU and DDP), and the results showed that INSL5 overexpression suppressed the sensitivity of NPC cells to the treatment of 5-FU or DDP (Fig EV4E and F). Furthermore, we detected the apoptosis pathway and found that INSL5 overexpression could suppress the cleavage of caspase 3, caspase 7, and caspase 9, indicating that INSL5 could suppress cell apoptosis (Fig EV4G). Additionally, the silencing of STAT5 in INSL5-overexpressing cells or JAK1 kinase inhibitor (Ruxolitinib) treatment could reverse the INSL5-induced glucose uptake (Fig 5H and I) and proliferation (Fig 5J). Collectively, these results suggest that the activation of STAT5 by INSL5 facilitated aerobic glycolysis, cell cycle progression, and apoptosis suppression may both contribute to NPC progression.

### Blocking of INSL5/GPCR142 can be a novel strategy for NPC therapy

To determine whether INSL5 could be used as a therapeutic target, we tested whether the monoclonal antibodies against INSL5 have neutralizing function. In the neutralization assay, we found that the INSL5 peptide contributed to NPC cell proliferation and invasion, while an anti-INSL5 monoclonal antibody (αINSL5, 50 and 100 μg/ml) could reverse the peptide-induced proliferation (Fig 6A) and invasion (Fig 6B) in a dose-dependent manner. Meanwhile, we detected

cell death and apoptosis after INSL5 antibody treatment and found that this antibody along had no obvious cytotoxicity (Fig EV5A). Additionally, when we use a GPCR142 antibody to block the binding of INSL5 and GPCR142, it also diminished INSL5 oncogenic function (Fig EV5B), which indicated that GPCR142 antibody can be potential therapeutic strategy for NPC. As we identified that INSL5 could induce glycolysis to promote cell proliferation, we then explored potential therapies using glycolysis inhibitors. Interestingly, the chemosensitivity assay demonstrated that INSL5 overexpression sensitizes NPC to the glycolysis inhibitor 2-DG (Fig 6C–E). Actually, conventional chemotherapy (cisplatin, carboplatin, 5-FU, or docetaxel) is the main strategy for NPC treatment. Previous results showed that INSL5 overexpression suppressed cell apoptosis during 5-FU treatment. Here, we found that 2-DG could reverse the chemoresistance, which indicated that 2-DG could sensitize INSL5 highly expressed NPC to chemotherapy (Figs 6F and EV5C). Finally, when we treated NPC patient-derived xenograft (PDX) tumor-bearing mice anti-INSL5 or GPCR142 neutralized antibodies, the tumor growth rate decreased, resulting in smaller tumor volume and lower tumor weight (Figs 6G and H and EV5D). Additionally, when we combined the treatment of blocking INSL5 or GPCR142 antibodies with conventional chemotherapy (DDP) in tumor-bearing mice, we found that INSL5 overexpression displayed chemoresistance to DDP treatment, which can be reversed by INSL5 or GPCR142 antibody treatment (Fig EV5E–G). Taken together, our results indicate that the overexpression of INSL5 promoted chemoresistance to conventional chemotherapy, while anti-INSL5 or GPCR142 neutralizing antibodies could suppress tumor alone or renders NPC cell sensitive to chemotherapy, revealing a novel strategy for NPC therapy.

## Discussion

The present study uncovered new roles of INSL5 in promoting cancer progression. Our results demonstrate that elevated plasma INSL5 levels can diagnostically distinguish NPC patients from normal individuals and predict poorer survival in NPC patients. Abnormally high expression of INSL5 could activate JAK1-STAT5 signaling to induce aerobic glycolysis to promote NPC tumor growth

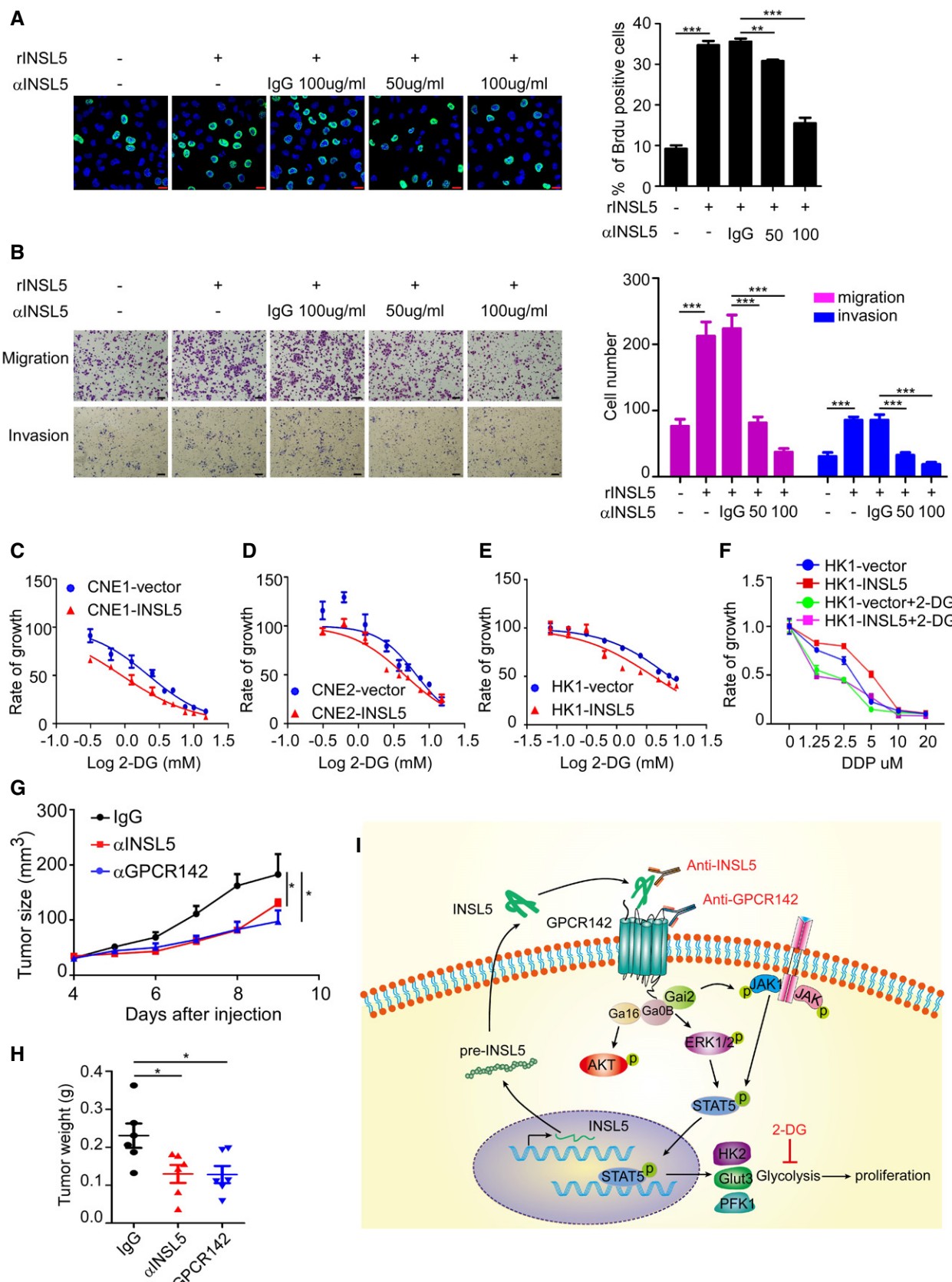

**Figure 6.**

◄

**Figure 6. Overexpression of INSL5 sensitizes NPC to a glycolysis inhibitor and INSL5 neutralized antibody.**

A Brdu assay for CNE1 cells stimulated with 100 ng/ml INSL5 peptide in the presence of an anti-INSL5 antibody at 50 or 100 µg/ml, or an isotype-matched IgG control (IgG) for 24 h. Scale bars represent 20 µm.

B Similar to (A), migration and invasion assay for CNE1 cells stimulated with 100 ng/ml INSL5 peptide in the presence of an anti-INSL5 antibody at 50 or 100 µg/ml, or an isotype-matched IgG control (IgG) for 24 h. Scale bars represent 100 µm.

C–E Growth curves of cell lines stably expressing INSL5 or vector 48 h post-treatment with the indicated dose of glycolysis inhibitor, 2-DG. CNE1 cells (C), CNE2 cells (D), and HK1 cells (E). $n = 4$ biological replicates for CNE1 and CNE2 cell lines, $n = 6$ biological replicates for HK1

F Growth curves of cell lines stably expressing INSL5 or vector 48 h post-treatment with the indicated dose of cis-platinum (DDP) and 2-DG. $n = 4$ biological replicates per group.

G, H Tumor size (G) and tumor weight (H) of nude mice burdened with NPC PDX tumor and intraperitoneally injected with anti-INSL5, anti-GPCR142, or isotype-matched IgG control (IgG) every 2 days. $n = 6$ mice per group.

I Working model for regulation of NPC progression and tumorigenesis by INSL5-GPCR142 axis. INSL5 is highly expressed in NPC tumor cells with poor prognosis in NPC patients. INSL5 can physically bind to the receptor GPCR142 to contribute to oncogenic function by reprogramming glycolysis, which is promoted by activation of the STAT5 signaling pathway. The anti-INSL5 neutralizing antibody, anti-GPCR142 neutralizing antibody, and glycolysis inhibitor could be attractive approaches for sensitizing INSL5-overexpressing NPC tumor cells.

Data information: In (A, B and F), data are presented as mean ± SD, in (C–E, G and H), data are presented as mean ± SEM, from three different experiments, and *P*-values were determined by unpaired t-test. \**P* < 0.05, \*\**P* < 0.01, \*\*\**P* < 0.001, ns, no significance. Exact *P*-values are specified in Appendix Table S4.

Source data are available online for this figure.

and invasion dependent on the INSL5 receptor GPCR142. Additionally, the acceleration of cell cycle progression and suppression of cell apoptosis by INSL5 may also contribute to NPC tumor progress. Finally, administration of the INSL5 and GPCR142 antagonist, the neutralizing antibodies, or the glycolysis inhibitor 2-DG markedly reduced cell proliferation, invasion, and chemoresistance. (Fig 6I).

Currently, the common biomarkers for NPC diagnosis and prognosis are mainly EBV antibodies (such as VCA-IgA, EA-IgA, and Zta-IgG) and circulating EBV DNA copy number (Liu *et al*, 2012; Kim *et al*, 2017). Among those biomarkers, VCA-IgA exhibits high sensitivity and specificity for the diagnosis of NPC, whereas VCA-IgA was undetectable in 4–24% of NPC patients, which might result in misdiagnosis (Chan *et al*, 2003). Similarly, regardless of the good prognostic prediction for NPC outcome, there are still 5–10% of NPC patients are negative for EBV DNA in plasma (Yip *et al*, 2014). In our study, we identified that NPC patients presented the highest INSL5 level in the plasma compared with normal individuals. Additionally, VCA-IgA-positive non-NPC individuals exhibited a medium INSL5 plasma level between the NPC cohort and VCA-IgA-negative normal individuals. It will be interesting to investigate whether those VCA-IgA-positive non-NPC individuals with higher plasma INSL5 have a highest risk to develop NPC. Inspiringly, INSL5 could also distinguish VCA-IgA-negative NPC patients from normal controls with a high sensitivity (90.2%) and specificity (85.9%). Thus, INSL5 could save as a diagnostic and prognostic marker complementary to EBV biomarkers.

Most cancer cells rely on aerobic glycolysis rather than respiration for their energy and metabolic needs, which is termed the "Warburg effect" and was described by Otto Warburg several decades ago (Schulze & Harris, 2012). In this way, cancer cells enhance glucose consumption by the glycolysis pathway for ATP production and biosynthetic metabolite generation, which are essential for the biomass production associated with cell growth (TeSlaa & Teitell, 2014; Boroughs & DeBerardinis, 2015; Wang *et al*, 2018). NPC is a specific disease with EBV infection. In EBV-infected NPC cells, the expression of EBV-encoded LMP1 has been reported to enhance glycolysis by modulating multiple signaling pathways, including those involving FGF1, AMPK, and mTORC1, to drive glucose metabolism (Chen *et al*, 2010; Ye *et al*, 2013; Zhang *et al*, 2017). Viral oncoproteins may also act as transcription factors to

upregulate the enzyme activities directly involved in glucose metabolism. INSL5 is a member of the relaxin/insulin superfamily and has a tertiary structure similar to that of insulin family members. Previous studies have mainly focused on the identification of the INSL5 receptor and the function of *Isnl5* in mice (Liu *et al*, 2005; Bathgate *et al*, 2013). Interestingly, in this study, we showed that INSL5 played a key role in reprogramming NPC cellular glucose metabolism to glycolysis to promote NPC progression, as demonstrated by *in vitro* functional assays and *in vivo* xenograft mouse models. It is reported that *Isnl5* promoted mouse appetite during conditions of energy deprivation in a $Insl5^{-/-}$ mice model (Grosse *et al*, 2014); here, we found INSL5 could enhance the "appetite" of NPC cells, which exhibited increased glucose uptake followed by increased production of ATP and other glycolytic intermediates. Previous studies have shown that gut microbiota can reduce colonic *Isnl5* expression and increase *Isnl5* expression in mice brain (Lee *et al*, 2016). This difference may be explained by the different energy demands of different tissues. Most strikingly, in our study, we found that EBV infection could transcriptionally induce INSL5 expression, but the specific mechanism needs further more studies. Collectively, most of NPC cancer cell lines express high levels of INSL5 despite many of these cell lines have lost EBV after maintaining *in vitro* for many passages. Thus, it will be interesting to delineate other mechanisms involved in INSL5 regulation. To date, there has been limited study on the function of INSL5 in cancer. Here, we found that INSL5 could promote NPC progression through its receptor, GPCR142. Mechanistically, we identified that INSL5 enhanced the phosphorylation and nuclear translocation of STAT5 and promoted glycolytic gene expression, which in turn induced metabolic reprogramming in cancer cells. A promising molecular target for the treatment of various cancers, STAT5 is activated in many cancers (Cotarla *et al*, 2004). STAT5 activation is induced by the phosphorylation of the key tyrosine residue in the STAT5 transactivation domain by growth factor receptors, Janus kinases (JAKs), SRC family kinases, and other tyrosine kinases (Levy & Darnell, 2002). This activation leads to dimerization, nuclear translocation, DNA binding, and the transcriptional induction of genes in the nucleus. It is reported that INSL5 could activate ERK1/2 and Akt kinases. Here, we found that STAT5 can be downstream of the INSL5-GPCR142 axis. The phosphorylation of Akt, ERK1/2, and

JAK1 increased after INSL5 treatment, different kinase inhibitor treatment revealed that JAK1 and ERK1/2 are involved in INSL5/GPCR142 downstream signaling cascade to activate STAT5, which could affect the phosphorylation status and nuclear distribution of STAT5, thus leading to the expression of glycolytic genes, such as HK2, Glut3, and PFK1.

Furthermore, the standard treatment for patients with locoregionally advanced NPC is radiotherapy combined with chemotherapy (Chen *et al*, 2012). It is worth noting that approximately 20% of NPC patients develop local recurrence after radiotherapy, and the relapsed NPC is usually more advanced (Al-Sarraf *et al*, 1998). Therefore, the development of novel strategies for NPC treatment should be of great importance. In this work, we found that the INSL5-GPCR142 axis can be a potential therapeutic target for NPC treatment. The neutralizing antibodies recognizing INSL5 and GPC142 can diminish cell proliferation and invasion *in vitro*. The nonlethal effects observed in INSL5 knockout mice (the knockout of INSL5 resulted in only a reduction in sperm motility and an irregular length estrous cycle) indicate that INSL5 may be good target for cancer therapy probably with no lethal effects. Consequently, we treated the INSL5-overexpressing NPC PDX mouse model with anti-INSL5 and anti-GPCR142 neutralizing antibodies, and as expected, these treatments markedly inhibited tumor growth. Furthermore, INSL5-overexpressing NPC tumor displayed chemoresistance to conventional chemotherapy, which can be reversed by glycolysis inhibitors 2-DG (Vander Heiden, 2011) or anti-INSL5/GPCR142 neutralizing antibodies. INSL5 also promoted the activation of STAT5, which inspired us to consider that a STAT5 inhibitor should be subjected to further testing for its possible use in NPC treatment.

In conclusion, we characterized INSL5 as a valuable biomarker for NPC diagnosis and prognosis. EBV infection contributes partially to the upregulation of INSL5 in NPC. INSL5 can physically bind to the receptor GPCR142 to activate JAK1 and ERK1/2 to enhance STAT5 phosphorylation and transcriptional activity in NPC cells. INSL5 exerts its oncogenic function partly by reprogramming glycolysis, which is promoted by activation of the STAT5 signaling pathway. The INSL5-GPCR142 axis can be a potential therapeutic target for NPC treatment. The anti-INSL5 neutralizing antibody, anti-GPCR142 neutralizing antibody, and glycolysis inhibitor could be attractive approaches for sensitizing INSL5-overexpressing NPC tumor cells.

# Materials and Methods

### Plasma and tissue samples

Plasma samples were collected from NPC patients at the time of diagnosis before treatment at the Sun Yat-sen University Cancer Center (SYSUCC) (Guangzhou, China) between 2009 and 2012. A total of 159 samples were collected from Zhejiang Cancer Hospital (Hangzhou, China) between 2012 and 2014, and 203 samples were collected from the People's Hospital of Zhongshan (Zhongshan, China). Plasma samples from healthy donors without diseases were collected by the physical examination department of the respective hospitals. NPC biopsy samples acquired from SYSUCC were employed to verify INSL5 expression with qRT–PCR. Twelve normal nasopharyngeal samples and 62 NPC biopsies were collected from histologically and clinically diagnosed NPC patients before clinical therapy between 2010 and

2011. The newly obtained NPC biopsies were immediately immersed in RNA later, stored overnight at 4°C, and then kept at −80°C until RNA extraction. This study was approved by the Institutional Research Ethics Committee of SYSUCC. Informed consent was obtained from all subjects, and the experiments conformed to the principles set out in the WMA Declaration of Helsinki and the Department of Health and Human Services Belmont Report.

### Cell culture and transfection

Normal immortalized NPEC cell lines (NP69, NPEC1-Bmi1, and NPEC2-Bmi1) were cultured in Keratinocyte serum-free medium (Invitrogen, Carlsbad CA USA). NPC cell lines (HK1, HNE1, CNE2, CNE1, SUNE1, and HONE1) were cultured in RPMI-1640 medium (Invitrogen, Carlsbad CA, USA) supplemented with 5% fetal bovine serum (FBS; HyClone, Logan, UT), and NPC cell line C666 was cultured in RPMI-1640 medium supplemented with 10% FBS. All the cells were transfected with plasmids using Lipofectamine 3000 (Invitrogen, Carlsbad, CA, USA) following the manufacturer's instructions.

### Plasmids and antibodies

The cDNAs of human INSL5 and GPCR142 were amplified by PCR and cloned into the pcDNA3.1 and pMSCV vectors with flag tags. The promoters of human INSL5, Glut3, HK2, and PFK1 were amplified by PCR and cloned into the pGL3-Basic vector. The following antibodies were used in this study: anti-INSL5 (Novus Biologicals, Littleton, CO, USA, NBP1-86343), anti-GPCR142 (Abmart, Shanghai, China, 24764-1hv-8/3H17), anti-Flag M2 (Sigma-Aldrich, St. Louis, MO, USA, F1804), anti-β-actin (ProteinTech Group Inc., Rosemont, IL, USA, 66009-1-1 g), anti-Enolase1 (Cell Signaling, 3810), anti-PFK1 (Abcam, ab154804), anti-HK2 (Cell Signaling, 2867), anti-Glut3 (Abcam, ab15311), anti-STAT5 (Cell Signaling, 25656 and 9363), anti-p-STAT5 (Cell Signaling, 9314 and 9351), anti-α-tublin (ProteinTech, 66031), anti-GAPDH (ProteinTech, 60004), anti-Lamin B1 (Abcam, ab16048), anti-p-JAK1 (upstate, 07-849), anti-p-Akt (Cell Signaling, 4060), anti-Akt (Cell Signaling, 9272), anti-p-ERK1/2 (Cell Signaling, 4377), anti-ERK1/2 (Cell Signaling, 4695), anti-p-STAT3 (Cell Signaling, 9145), anti-cyclin B (Santa Cruz, sc-7393), anti-cyclin E (upstate, 05-363), anti-cyclin D (Cell Signaling, 2978), anti-p27 (Cell Signaling, 3686), anti-c-myc (Cell Signaling, 5605), anti-BCL2 (Cell Signaling, 15071), anti-BCL-xL (Cell Signaling, 2764), anti-cleaved caspase 3 (Cell Signaling, 9664), anti-caspase 3 (Cell Signaling, 9662), anti-cleaved caspase 7 (Cell Signaling, 9491), anti-caspase 7 (Cell Signaling, 9492), anti-cleaved caspase 9 (Cell Signaling, 9501), anti-caspase 9 (Cell Signaling, 9502), anti-BrdU (Abcam, ab6326), and FITC-conjugated anti-rabbit IgG (Invitrogen, Carlsbad, CA, USA). The antibodies were used at a 1:1,000 dilution for immunoblotting experiments and 1:200 dilution for immunofluorescence. Anti-INSL5 and anti-GPCR142 antibodies were used at 50 μg/μl or 100 μg/μl for the antibody blocking assay. The glycolysis inhibitor 2-DG was purchased from Sigma-Aldrich (D8375).

### ELISA detection of INSL5 in the plasma

A purified monoclonal anti-INSL5 antibody (46B8, 1 mg/ml) was diluted to 2 μg/ml in 10 mM PBS at pH 7.4. Microplates (96-well,

Costar, #42592) were coated with 200 µl of the diluted antibody at room temperature overnight. After incubation, the plates were washed with 0.05% PBS containing 0.05% Tween-20 (PBST), and the unoccupied sites were blocked with 5% BSA in PBS. Then, the plates were washed three times with PBST. After washing, 100 µl of each plasma sample (1:10 diluted in PBS) and the standards (INSL5 short A-&B-chains, Phoenix Biotech, #035-70, diluted to 2,000, 1,000, 500, 250, 125, 62.5, 32.25, and 0 pg/µl) was added and incubated at 37°C for 2 h. After three washes, 100 µl of detection antibody (anti-INSL5 antibody, 23G9, biotinylated by an EZ-Link Sulfo-NHS-LC-Biotin kit, Thermo Fisher Scientific, #21327, diluted to 45 ng/ml) was added, and the plates were incubated at 37°C for 2 h. After three washes, 100 µl of HRP-labeled streptavidin (R&D Systems, DY998, diluted 1:200 with PBS) was added, and the plates were incubated at room temperature for 20 min. After that, the plates were washed five times before adding a tetramethylbenzidine reagent (Sigma-Aldrich, St Louis, MO, USA) for 10 min at 37°C. The reaction was stopped with 2 M $H_2SO_4$, and the optical density (OD) value at 450 nm was determined using a microplate reader. Finally, the concentration of INSL5 in the samples was calculated according to the standards. The ROC curve was plotted to identify the optimum cut-off value of plasma INSL5 level for diagnosis and prognosis.

### siRNA transfection and qRT–PCR

The siRNAs targeting INSL5 were designed as siINSL5-#1 (UCUGUGGGCUAGAAUACAU) and siINSL5-#2 (GUCAUCUAUAU CUGUGCUA). The siRNAs targeting GPCR142 were designed as siGPCR142-#1 (UCCGAUGCCUGUCAAAUUC) and siGPCR142-#2 (UCCUGGUGGCUUCCUUCUU). The siRNAs targeting STAT5 were designed as siSTAT5-#1 (UGAUGGAGGUGUUGAAGAA) and siST AT5-#2 (GCAAUGAGCUUGUGUUCCA). The negative control (siNC) was nonhomologous to any human genome sequence and purchased from RiboBio Co., Ltd. Predetermined cells were plated in 6-well plates for 12 h and then transfected with 20 nM siRNA mixed with 5 µl of Lipofectamine RNAiMAX (Invitrogen) according to the manufacturer's instructions. At 36–48 h after transfection, the cells were harvested for further experimentation. Total RNA from the cells was extracted with TRIzol reagent following the manufacturer's instructions (qRT–PCR primers are attached in Appendix Table S3).

### MTT assay

An MTT assay (Sigma, St Louis, MO, USA) was used to measure cell growth according to the manufacturer's instructions. Briefly, about 1,500 cells were seeded in 96-well plates and cultured in RPMI1640 supplemented with 5% fetal bovine serum. Cell proliferation reflected in OD490 nm was examined on 1, 2, 3, 4, and 5 days followed the standard procedures. All experiments were repeated three times.

### Colony formation assay

Nasopharyngeal carcinoma cell lines after knockdown or overexpression of INSL5 were plated in triplicate in 6-well plates at 400 cells per well and cultured for 10 days. After fixation with methanol for 15 min, the colonies were stained with 0.5% crystal violet in 20% methanol for 15 min and counted. All experiments were repeated three times.

### BrdU incorporation

The BrdU incorporation assay was performed as described previously (Zeng et al, 2012). Briefly, INSL5-overexpressing cell lines or siGPCR142-knockdown cells ($5 \times 10^4$) were seeded on coverslips in a 24-well plate. After 24 h, the cells were incubated with BrdU (20 µM) for 2 h and then stained with an anti-BrdU (Abcam) primary antibody and a FITC-conjugated anti-rat IgG secondary antibody according to the manufacturer's instructions. The cells were examined under a confocal laser-scanning microscope. Green indicates positive BrdU incorporation, and blue staining indicates the nuclei.

### Transwell assay

Nasopharyngeal carcinoma cell lines ($4–5 \times 10^4$ cells for migration assay and $6–7.5 \times 10^4$ cells for invasion assay) with INSL5 knockdown or overexpression resuspended in 200 µl FBS-free RPMI 1640 were seeded in the top chamber of the Transwell (for invasion assay, the chamber was coated with Matrigel). The lower chamber was filled with 500 µl 10% FBS RPMI 1640 as a chemo-attractant. After incubation for 24 h (36–48 h for invasion assay), cells on the lower surface of the member were fixed, stained, and counted. Each group of cells was done in triplicate.

### Immunofluorescence

Cells were stained for immunofluorescence on coverslips as described previously (Kong et al, 2010). Briefly, cells that received the indicated treatment were fixed in 4% paraformaldehyde, permeabilized in 0.1% Triton X-100, and blocked with 5% BSA in PBS. Then, the cells were incubated with primary antibodies overnight at 4°C, followed by incubation with FITC- or rhodamine-conjugated secondary antibodies at room temperature in the dark for 1 h. The cells were counterstained with DAPI for 5 min. Images were captured with a confocal laser-scanning microscope (Olympus FV1000).

### Cell cycle and cell apoptosis analysis

Cell cycle analysis was performed by propidium iodide (PI) staining according to the manufacture's instruction and analyzed by a Gallios flow cytometer. Drug-induced apoptosis (5-FU or DDP) was evaluated by Annexin V and propidium iodide (PI) staining using an Annexin V Alexa Fluor 647/PI Apoptosis detection Kit (4A Biotech Co., Ltd., #FXP023).

### Dual-luciferase assay

Luciferase activity was measured using the Dual-Luciferase Reporter Assay System (Promega). 293T cells were transfected with the indicated plasmids for 36 h, and the firefly luciferase activity was assayed and normalized to that of Renilla luciferase according to the manufacturer's instructions.

## ChIP assay

The ChIP assay was performed as previously described (Song *et al*, 2009). Briefly, $2 \times 10^6$ cells that received the indicated treatment were plated per 100-mm-diameter dish and treated with formaldehyde to cross-link chromatin-associated proteins and DNA. The cells were scraped in PBS and resuspended in lysis buffer, and the nuclei were isolated and sonicated to shear the DNA to 500 bp–1 kb fragments (verified by agarose gel electrophoresis). Equal aliquots of chromatin supernatants were subjected to overnight immunoprecipitation (IP) with different antibodies as indicated, and anti-myc or IgG antibodies were used as negative controls. DNA was extracted with phenol–chloroform, and the indicated regions were amplified by qPCR (qRT–PCR primers are attached in Appendix Table S3).

## Metabolic profiling

Metabolites were analyzed by liquid chromatography-mass spectrometry (LC-MS), as previously described (Lewis *et al*, 2008). Briefly, vector cells and INSL5 overexpressed cells were cultured in low-glucose nutrient conditions for 24 h in 10-cm dish until 80% confluence and changed the supernatant with fresh medium 2 h before cell harvest. After that, pull out the culture medium and wash the cells with 37°C warmed PBS for twice. The cells were fixed with 80% methanol at −80°C for 2 h and scraped cells. After centrifuging at 14,000 *g* for 20 min, the supernatant was dried into a powder and resolved in 80% methanol according to the total protein concentration. After complete dissolution, the solutions were analyzed by LC-MS.

## Cellular oxygen consumption and extracellular acidification rate

Cellular extracellular acidification rate (ECAR) and oxygen consumption rate (OCR) were, respectively, measured by a Seahorse XF96 Extracellular Flux Analyzer (Agilent Technologies, Santa Clara, CA, USA). ECAR was detected with XF Glycolysis stress Test Kit (Seahorse Bioscience, #103020-100) according to the manufacturer's instructions. ORC was determined with XF Cell Mito Stress Test Kit (Seahorse Bioscience, #103015-100) according to the manufacturer's instructions.

## Measurement of glucose uptake and lactate production

The glucose uptake assay was performed as previously published. Briefly, cells that received the indicated treatments were plated in 6-well plates, and then, 10 μM 2-NBDG (Invitrogen) was added to the medium for 2 h, when the cells were 80% confluent. After incubation, flow cytometry was performed to detect the percentage of glucose uptake. Lactate production, ATP concentration, and HK2 activity were determined with specific kits (BioVision) according to the manufacturer's instructions.

## Subcellular fractionation

Cells were harvested with ice-cold PBS, and then, the cytosolic fraction and nuclear fraction were separated and extracted according to the manufacturer's instructions for the Nuclear and Cytoplasmic Protein Extraction Kit (Beyotime, cat# P0028). The subcellular

**The paper explained**

**Problem**

Nasopharyngeal carcinoma (NPC) is a malignant epithelial tumor with a unique geographic distribution, as it is mainly found in Southern China and South-East Asia. Metabolic reprogramming plays important roles in development and progression of NPC, but the potential key molecular targets involved in modulating NPC metabolism remain to be identified and the underlying mechanism has not been completely defined. Additionally, the specific functions of INSL5 in tumors remain to be elucidated.

**Results**

Our study characterized INSL5 as a valuable biomarker for NPC diagnosis and prognosis. EBV infection induces increased expression of INSL5 in NPC. INSL5 can physically bind to the receptor GPCR142 to activate JAK1 and ERK1/2 to enhance STAT5 phosphorylation and transcriptional activity in NPC cells. INSL5 exerts its oncogenic function by reprogramming glycolysis, which is promoted by activation of the STAT5 signaling pathway. The INSL5-GPCR142 axis can be a potential therapeutic target for NPC treatment. The anti-INSL5 neutralizing antibody, anti-GPCR142 neutralizing antibody, and glycolysis inhibitor could be attractive therapeutic approaches for INSL5-overexpressing NPC.

**Impact**

These results shed light on the mechanism of INSL5 in mediating glucose metabolism reprogramming and tumor promotion in NPC and highlight a novel biomarker for NPC diagnosis, prognosis, and targeted therapy.

fractions were boiled with SDS–PAGE loading buffer for Western blot analysis.

## *In vivo* tumorigenesis and PDX treatment

Female Nu/Nu nude mice (5–6 weeks old) were purchased from Beijing Vital River Laboratory Animal Technology Co., Ltd. All studies were approved and supervised by the Animal Ethics Committee of SYSUCC and performed in accordance with the relevant guidelines and regulations. To determine the effect of INSL5 on NPC growth, CNE2 cells ($1 \times 10^5$ stably overexpressing the control or INSL5) and HK1 cells ($4 \times 10^6$ stably overexpressing the control or INSL5) in 100 μl of RPMI-1640 medium containing 20% Matrigel were injected subcutaneously into mice. One week after cell injection, the length and width of the tumors were measured by Vernier calipers every other day. The mice were euthanized before they met the institutional euthanasia criteria for tumor burden and overall health condition. Then, the primary tumors were collected and weighed. For conventional chemotherapy, HK1 cells ($2 \times 10^6$ stably overexpressing the control or INSL5) in 100 μl of RPMI-1640 medium containing 20% Matrigel were injected subcutaneously into mice. Five days after cell injection, we treated the mice with cisplatinum (DDP) 4 mg/kg every 3 days and mouse IgG, anti-INSL5 antibody (200 μg each), or anti-GPCR142 antibody (200 μg each) every 2 days. For PDX experiments, we divided the tumor sample from an NPC PDX into equal pieces and then implanted the pieces in other normal Nu/Nu nude mice (5–6 weeks old) with random grouping. Three days after implantation, we treated the mice with

an anti-INSL5 antibody (250 μg each) or anti-GPCR142 antibody (250 μg each) via intraperitoneal injection.

### Statistical analysis

All statistical analyses were conducted using GraphPad Prism, version 5.0 or 8.0 (San Diego, USA). Grouped data are presented as mean ± SD or mean ± SEM unless otherwise stated. Differences between two groups were assessed using a two-tailed unpaired t-test. Survival outcomes were compared using the Kaplan–Meier method and the log-rank test. Significance was determined at $P < 0.05$.

## Data availability

Primary datasets have been generated and deposited in the Research Data Deposit (RDD) public platform (http://www.researchdata.org.cn), with the approval RDD number as RDDB2020000874.

**Expanded View** for this article is available online.

## Acknowledgements

This work was financially supported by the National Key Research and Development Program of China (2017YFC0908503, 2017YFA0505600, 2016YFA0502101), National Natural Science Foundation of China (81830090, 81520108022, 81621004, 81802775, 81672703, and 81772883), Natural Science Foundation of Guangdong Province (No.2017A030312003), Guang Dong Province Key Research and Development program (2019B020226002), the Guangzhou Science Technology and Innovation Commission (201607020038), and China Postdoctoral Science Foundation (2019M663252).

## Author contributions

M-SZ conceived and designed the experiments, provided supervision and wrote the manuscript. S-BL conceived the experiments. S-BL, Y-YL, and LY performed, analyzed the key experiments, and wrote the manuscript. AZ, H-YL, L-QT, S-GF, HZ, SX, M-ZL, and QZ performed the experiments designed and analyzed the data. S-JL, W-LL, PH, Y-XZ, M-FJ, Y-MZ, Z-QL, and J-HS contributed reagents and materials.

## Conflict of interest

The authors declare that they have no conflict of interest.

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
