## [Review Process File · EMBO Molecular Medicine]

Autocrine INSL5 promotes tumor progression and glycolysis via activation of STAT5 signaling

Shi-Bing Li, Yan-Yan Liu, Li Yuan, Ming-Fang Ji, Ao Zhang, Hui-Yu Li, Lin-Quan Tang, Shuo-Gui Fang, Hua Zhang, Shan Xing, Man-Zhi Li, Qian Zhong, Shao-Jun Lin, Wan-Li Liu, Peng Huang, Yi-xin Zeng, Yu-Ming Zheng, Zhi-Qiang Ling, Jian-Hua Sui, and Mu-Sheng Zeng

DOI: [10.15252/emmm.202012050](https://doi.org/10.15252/emmm.202012050)

Corresponding author(s): Mu-Sheng Zeng (zengmsh@sysucc.org.cn)

Review Timeline:

Submission Date:	18th Jan 20
Editorial Decision:	4th Feb 20
Revision Received:	14th May 20
Editorial Decision:	25th May 20
Revision Received:	15th Jun 20
Accepted:	18th Jun 20

Editor: Jingyi Hou

Transaction Report:

4th Feb 2020

Dear Prof. Zeng,

Thank you for the submission of your manuscript to EMBO Molecular Medicine. We have now received feedback from the two referees whom we asked to evaluate your manuscript.

As you will see from the reports below, both referees acknowledge the potential interest of the study. However, they also raise a number of concerns about your work, which should be convincingly addressed in a major revision of the present manuscript. In particular, additional experiments are required to strengthen the mechanisms underlying the pro-oncogenic role of INSL5 and to make the study more conclusive. Referee #1 also requested data on the INSL5 knock-out mice, which we would encourage you to provide.

All other issues raised by the referees need to be satisfactorily addressed as well. We would welcome the submission of a revised version within three months. Please note that EMBO Molecular Medicine strongly supports a single round of revision and that, as acceptance or rejection of the manuscript will depend on another round of review, your responses should be as complete as possible.

Please also contact us as soon as possible if similar work is published elsewhere. If other work is published, we may not be able to extend the revision period beyond three months.

I look forward to receiving your revised manuscript.

Yours sincerely,
Jingyi Hou

Jingyi Hou
Editor
EMBO Molecular Medicine

*** Instructions to submit your revised manuscript ***

**** PLEASE NOTE **** As part of the EMBO Publications transparent editorial process initiative (see our Editorial at <https://www.embopress.org/doi/pdf/10.1002/emmm.201000094>), EMBO Molecular Medicine will publish online a Review Process File to accompany accepted manuscripts.

To submit your manuscript, please follow this link:

Link Not Available

- 1) a .doc formatted version of the manuscript text (including Figure legends and tables). Please make sure that the changes are highlighted to be clearly visible to referees and editors alike.
- 2) separate figure files*
- 3) supplemental information as Expanded View and/or Appendix. Please carefully check the authors guidelines for formatting Expanded view and Appendix figures and tables at <https://www.embopress.org/page/journal/17574684/authorguide#expandedview>
- 4) a letter INCLUDING the reviewers' reports and your detailed responses to their comments (as Word file)

Also, and to save some time should your paper be accepted, please read below for additional information regarding some features of our research articles:

- 5) The paper explained: EMBO Molecular Medicine articles are accompanied by a summary of the articles to emphasize the major findings in the paper and their medical implications for the non-specialist reader. Please provide a draft summary of your article highlighting
 - the medical issue you are addressing,
 - the results obtained and
 - their clinical impact.

- 6) For more information: There is space at the end of each article to list relevant web links for further consultation by our readers. Could you identify some relevant ones and provide such information as well? Some examples are patient associations, relevant databases,

OMIM/proteins/genes links, author's websites, etc...

7) Author contributions: the contribution of every author must be detailed in a separate section (before the acknowledgments).

8) EMBO Molecular Medicine now requires a complete author checklist (<https://www.embopress.org/page/journal/17574684/authorguide>) to be submitted with all revised manuscripts. Please use the checklist as a guideline for the sort of information we need WITHIN the manuscript as well as in the checklist. This is particularly important for animal reporting, antibody dilutions (missing) and exact p-values and n that should be indicated instead of a range.

9) Every published paper now includes a 'Synopsis' to further enhance discoverability. Synopses are displayed on the journal webpage and are freely accessible to all readers. They include a short stand first (maximum of 300 characters, including space) as well as 2-5 one sentence bullet points that summarise the paper. Please write the bullet points to summarise the key NEW findings. They should be designed to be complementary to the abstract - i.e. not repeat the same text. We encourage inclusion of key acronyms and quantitative information (maximum of 30 words / bullet point). Please use the passive voice. Please attach these in a separate file or send them by email, we will incorporate them accordingly.

You are also welcome to suggest a striking image or visual abstract to illustrate your article. If you do please provide a jpeg file 550 px-wide x 400-px high.

10) A Conflict of Interest statement should be provided in the main text

11) Please note that we now mandate that all corresponding authors list an ORCID digital identifier. This takes <90 seconds to complete. We encourage all authors to supply an ORCID identifier, which will be linked to their name for unambiguous name identification.

Currently, our records indicate that there is no ORCID associated with your account.

Please click the link below to provide an ORCID:

Link Not Available

12) The system will prompt you to fill in your funding and payment information. This will allow Wiley to send you a quote for the article processing charge (APC) in case of acceptance. This quote takes into account any reduction or fee waivers that you may be eligible for. Authors do not need to pay any fees before their manuscript is accepted and transferred to our publisher.

Photos 400-800 DPI

Figures are not edited by the production team. All lettering should be the same size and style; figure

panels should be indicated by capital letters (A, B, C etc). Gridlines are not allowed except for log plots. Figures should be numbered in the order of their appearance in the text with Arabic numerals. Each Figure must have a separate legend and a caption is needed for each panel.

*Additional important information regarding figures and illustrations can be found at <http://bit.ly/EMBOPressFigurePreparationGuideline>

***** Reviewer's comments *****

Referee #1 (Comments on Novelty/Model System for Author):

Needs additional experiments as detailed in the report

Referee #1 (Remarks for Author):

The manuscript by Li and colleagues is an interesting piece of work convincingly showing the potential of the insulin-like peptide 5 (INSL5) as biomarker of diagnosis and of bad prognosis of patients bearing nasopharyngeal carcinomas (NPC). Less developed are the aspects related with the mechanisms that underlie the pro-oncogenic role of INSL5. In this regard, they overexpressed INSL5 or silenced its high-affinity receptor, the G-protein coupled receptor RXFP4 (GPCR142), in different cellular lines to demonstrate that INSL5, or its downstream signaling, favor in vitro proliferation and invasion of the cells and in vivo tumor growth in xenograft mouse models. In the search for a mechanism of action of INSL5, they report that both mRNAs, enzymes and metabolites of the glycolytic pathway are significantly increased upon the action or signaling through INSL5, whereas oxidative phosphorylation and metabolites of the TCA cycle negatively correlate with the activity of INSL5. These and some other additional results led the authors to suggest that INSL5 mechanism of action is through transcriptional upregulation of glycolysis. Mechanistically, they report that INSL5 triggers the phosphorylation and nuclear translocation of transcription factor STAT5, which seems to interact with the promoters of some glycolytic genes and increase their activity upon INSL5 stimulation. Finally, they show that antibodies against INSL5 or its receptor GPCR142 diminish the pro-oncogenic activity of INSL5 in patient-derived tumor xenografts suggesting that they could provide targets for the treatment of NPC. Overall, the paper needs of additional experiments to convey a convincing message.

Major points

1. INSL5- and RXFP4-mediated metabolic reprogramming are poorly characterized and their effects on mitochondrial respiration and oxidative phosphorylation need to be reported. Is the cellular mitochondrial content affected?
2. The two-fold increase in cellular ATP content in response to the overexpression of INSL5 (Fig. 4I) or its sharp decrease in knock-down cells (Fig. 4K) is difficult to reconcile with the marginal changes published in cellular ATP content. What is the effect of 2DG and of oligomycin in cellular ATP content?
3. The phosphorylation and nuclear translocation experiments of STAT5 in response to INSL5 (Fig. 5B-D) are not convincing. They require quantification and the incorporation of the results obtained in histograms in the same figure. STAT5 is known to activate anti-apoptotic genes. Is any of these anti-apoptotic genes activated in response to INSL5? What is the contribution of preventing cell death in tumor accretion mediated by INSL5 action?
4. The INSL5/RXFP4 downstream signaling cascade to STAT5 phosphorylation needs to be delineated. Which kinase (ERK, JAK, Akt...) is involved?
5. Apparently, blocking glycolysis with 2DG has no relevant effect in tumor growth (Fig. 6F), raising

doubts about the relevance of glycolytic reprogramming in the INSL5-mediated pro-oncogenic role. The authors should consider combining the treatment of blocking INSL5 or GPCR142 antibodies with conventional chemotherapy (cisplatin, carboplatin, 5FU or docetaxel) regimens of NPC.

6. Is the INSL5 knock-out mice resistant to the growth of NPC?

Minor points

There are some spelling errors that need amendment in the text (line 131, OASL mRNA....) and in the figures (Fig. 4K, reads Akt when it should be ATP)

Explain why the 3.73 cut-off value has been selected (line 185) in KM analyses

The mouse genes need to be appropriately quoted.

Referee #2 (Comments on Novelty/Model System for Author):

Under my knowledge, INSL5 has not been before involved in the progression of NPC. Moreover, it is shown in this study that INSL5 can be a prognosis and diagnosis marker of this cancer type. This opens the door also to analyze this protein other cancers.

Referee #2 (Remarks for Author):

Li et al. describe in this paper a new role for INSL5 in the progression of nasopharyngeal carcinoma. They start by proving the use of INSL5 quantification as a diagnostic and prognostic marker of NPC in human patients. Next, they use cellular models of NPC to analyze the underlying mechanisms by which INSL5 regulates the proliferation and growth of these cancer cells. They show, using different approaches, that INSL5 through binding to its receptor GPCR142, activates the STAT5 pathway to regulate the expression of genes involved in the control of glycolysis. They claim that through this process, INSL5 facilitates the metabolic switch required for cancer cells to proliferate. The study is very extensive, and covers from studies in human patients of cancer to work in mouse models. Furthermore, through the use of cellular models they unravel the molecular mechanisms underlying the effects of INSL5 on cancer cells. This is a novel and original work that has been well conducted. There is no doubt about the relevance of this study for cancer research and treatment. Some additional work, as is next suggested, would further improve the quality of the manuscript and reinforce the conclusions.

1. The authors claim that the function of INSL5 in the proliferation and growth of NPC cancer cells is to promote the required metabolic switch. While this is well proven in this study, the authors cannot conclude that this is the only, or the most important mechanisms that underlie the effects of INSL5 in cancer progression. Indeed, in the figure 6F, the growth of the tumors is only minimally changed when glycolysis is inhibited by 2-DG.

2. The STAT5 pathway has been involved in tumorigenesis in several types of cancer. Previous studies show that STAT5 can mediate oncogenic signals and regulate cell cycle progression, proliferation and promote cancer cell survival. Are there other STAT5-regulated pathways, in addition to glycolysis, changed in the models used?

3. Are JAK kinases, the upstream activators of the STAT pathway also involved? Are JAK kinase inhibitors or STAT5 knock-down experiments abrogate the effects of INSL5?

4. Several reports suggest that there is an important correlation between the JAK/STAT pathway and cell metabolism, notably the activation of hypoxia-inducible factors and the alteration of mitochondrial activity. In addition to the regulation of glycolysis genes, STAT5 has also been involved in mitochondrial function. A more detailed functional evaluation of mitochondria and glycolytic function of the cells with or without INSL5 (using for instance Seahorse analysis) would further prove the direct involvement of this protein in the control of metabolism in these cancer cells.

5. In the Figure 6 A-B, the authors show the effects of INSL5 inhibition by using a monoclonal antibody. Cell death and apoptosis should be analyzed in these cells.

Point-by-point response

We thank all the reviewers for their constructive comments and suggestions. We have performed additional experiments and revised the manuscript accordingly. We believe that we have addressed all the concerns raised by reviewers in the revised manuscript. All changes in the revised manuscript are highlighted in yellow for your attention. The following is a point-by-point response on how we revised our manuscript.

Referee #1 (Comments on Novelty/Model System for Author):

Needs additional experiments as detailed in the report

Referee #1 (Remarks for Author):

The manuscript by Li and colleagues is an interesting piece of work convincingly showing the potential of the insulin-like peptide 5 (INSL5) as biomarker of diagnosis and of bad prognosis of patients bearing nasopharyngeal carcinomas (NPC). Less developed are the aspects related with the mechanisms that underlie the pro-oncogenic role of INSL5. In this regard, they overexpressed INSL5 or silenced its high-affinity receptor, the G-protein coupled receptor RXFP4 (GPCR142), in different cellular lines to demonstrate that INSL5, or its downstream signaling, favor in vitro proliferation and invasion of the cells and in vivo tumor growth in xenograft mouse models. In the search for a mechanism of action of INSL5, they report that both mRNAs, enzymes and metabolites of the glycolytic pathway are significantly increased upon the action or signaling through INSL5, whereas oxidative phosphorylation and metabolites of the TCA cycle negatively correlate with the activity of INSL5. These and some other additional results led the authors to suggest that INSL5 mechanism of action is through transcriptional upregulation of glycolysis. Mechanistically, they report that INSL5 triggers the phosphorylation and nuclear translocation of transcription factor STAT5, which seems to interact with the promoters of some glycolytic genes and increase their activity upon INSL5 stimulation. Finally,

they show that antibodies against INSL5 or its receptor GPCR142 diminish the pro-oncogenic activity of INSL5 in patient-derived tumor xenografts suggesting that they could provide targets for the treatment of NPC. Overall, the paper needs of additional experiments to convey a convincing message.

Major points

1. INSL5- and RXFP4-mediated metabolic reprogramming are poorly characterized and their effects on mitochondrial respiration and oxidative phosphorylation need to be reported. Is the cellular mitochondrial content affected?

Response:

Thank you very much. This is a great point. In the revised manuscript, we have performed Seahorse analysis in three different INSL5 overexpressing cell lines (CNE1, CNE2 and HK1) to detect extracellular acidification rate (ECAR) and oxygen consumption rate (OCR) using a XF96 Extracellular Flux analyzer. We demonstrated that INSL5 overexpression significantly increased glycolysis (Fig 4F-G and Fig S5G) and impaired oxidative phosphorylation (OXPHOS) (Fig S5H), which indicated that INSL5 overexpression could promote metabolism shift from OXPHOS to aerobic glycolysis.

2. The two-fold increase in cellular ATP content in response to the overexpression of INSL5 (Fig. 4I) or its sharp decrease in knock-down cells (Fig. 4K) is difficult to reconcile with the marginal changes published in cellular ATP content. What is the effect of 2DG and of oligomycin in cellular ATP content?

Response:

Thank you very much for your suggestion. We are sorry that we made a mistake before, the ATP concentration should be normalized with protein concentration according to the manufacturer's instructions, but we didn't. In the revised manuscript we detected the ATP level in INSL5 overexpressing cell lines and knock-down cells, we got the similar results after protein concentration normalization. CNE1, CNE2 and

HK1 cell respectively showed about 34%, 22%, and 26% increase of ATP level after INSL5 overexpression. INSL5 knockdown in CNE2-EBV and HNE1-EBV cells respectively showed about 35% and 16% decrease of ATP level. We treated the INSL5 overexpressing cells with oligomycin or 2-DG, we found that oligomycin treatment slightly increased the ATP level, while 2-DG treatment greatly decreased the ATP level, INSL5 overexpressing cells especially displayed a greater extent reduction than control cells, which indicated that INSL5 overexpression indeed promoted glucose metabolism shift from OXPHOS to glycolysis (Fig 4J and Fig S5K).

3. The phosphorylation and nuclear translocation experiments of STAT5 in response to INSL5 (Fig. 5B-D) are not convincing. They require quantification and the incorporation of the results obtained in histograms in the same figure. STAT5 is known to activate anti-apoptotic genes. Is any of these anti-apoptotic genes activated in response to INSL5? What is the contribution of preventing cell death in tumor accretion mediated by INSL5 action?

Response:

Thanks for your helpful suggestion. For Fig.5B-D, we quantified the western blotting results by Image J and labeled the fold change just above the indicated bands. We agreed with you that STAT5 is known to activate anti-apoptotic genes, like c-myc, BCL2 and BCL-xL. We detected those genes in INSL5 overexpressing and control cells, and found that INSL5 overexpression only increased c-myc expression, not BCL2 and BCL-xL (Fig S6D). We also detected cell apoptosis under conventional chemotherapy (5-FU and DDP), the results showed that INSL5 overexpression suppressed the sensitivity of NPC cells to 5-FU or DDP. Furthermore, we detected the apoptosis pathway, and found that INSL5 overexpression could suppress the cleavage of caspase 3, caspase7 and caspase 9. Taken together, all of those data suggested that INSL5 overexpression could promote chemoresistance to 5-FU or DDP via inhibiting cell apoptosis (Fig S6E-G).

4. The INSL5/RXFP4 downstream signaling cascade to STAT5 phosphorylation needs to be delineated. Which kinase (ERK, JAK, Akt...) is involved?

Response:

Thank you, this is an excellent question. As reported INSL5 could activate ERK and Akt, we detected the activation of ERK, Akt and JAK1, found that INSL5 could promote the activation of ERK, Akt and JAK1. Meanwhile, we treated the cell with different kinase inhibitor to detect the phosphorylation of STAT5 and found that JAK1 (Ruxolitinib) and ERK1/2 (U0126) inhibitors, not Akt inhibitor (MK2206), could reverse STAT5 activation. Taken together, those results indicated that JAK and ERK contributed to STAT5 activation induced by INSL5/RXFP4 (Fig 5E).

5. Apparently, blocking glycolysis with 2DG has no relevant effect in tumor growth (Fig. 6F), raising doubts about the relevance of glycolytic reprogramming in the INSL5-mediated pro-oncogenic role. The authors should consider combining the treatment of blocking INSL5 or GPCR142 antibodies with conventional chemotherapy (cisplatin, carboplatin, 5FU or docetaxel) regimens of NPC.

Response:

Thanks for your great suggestion. We agree that glycolysis inhibitor 2-DG only shows minor effect in tumor growth. As suggested, we examine whether 2-DG could reverse INSL5 enhanced chemoresistance. We found that 2-DG could sensitize INSL5 highly expressed NPC to chemotherapy (Fig 6F and Fig S7C). Additionally, we combined the treatment of blocking INSL5 or GPCR142 antibodies with conventional chemotherapy (DDP) in tumor-bearing mice, and we found that INSL5 overexpression displayed chemoresistance to DDP treatment, which can be reversed by INSL5 or GPCR142 antibodies treatment (Fig S7E-G). Taken together, those results indicated that Anti-INSL5/GPCR142 neutralized antibodies could diminish the chemoresistance induced by INSL5 overexpression

6. Is the INSL5 knock-out mice resistant to the growth of NPC?

Response:

We are very grateful for your nice suggestion. Indeed, transgenic INSL5 knock-out or overexpressing mice will make our conclusions stronger. But there is no ideal animal model for NPC and we also can not get the transgenic mice in the limited time, we will follow up this exploration as your suggestion in the future. Anyway, thank you very much for your suggestion.

Minor points

There are some spelling errors that need amendment in the text (line 131, OASL mRNA....) and in the figures (Fig. 4K, reads Akt when it should be ATP)

Response:

We have corrected those mistakes in the revised manuscript as suggested. Thank you very much for the careful correction, which help us avoid these unthoughtful mistakes.

Explain why the 3.73 cut-off value has been selected (line 185) in KM analyses

Response:

Thank you. We mentioned this in the methods (line 557). According to many reported methods, we performed receiver operating characteristic (ROC) curve analysis to identify the optimum cut-off value of plasma INSL5 level for prognosis. We divided all the NPC patients into two groups, progression group and progression free group, then we draw the ROC curve to predict tumor progression and calculated the Yuden index. Finally, the cut-off value for INSL5 was defined as the value with the maximization of Yuden index.

The mouse genes need to be appropriately quoted.

Response:

Thank you for your reminding. We have corrected all the mouse genes to italics and lowercase letters to distinguish from human genes.

Referee #2 (Comments on Novelty/Model System for Author):

Under my knowledge, INSL5 has not been before involved in the progression of NPC. Moreover, it is shown in this study that INSL5 can be a prognosis and diagnosis marker of this cancer type. This opens the door also to analyze this protein other cancers.

Referee #2 (Remarks for Author):

Li et al. describe in this paper a new role for INSL5 in the progression of nasopharyngeal carcinoma. They start by proving the use of INSL5 quantification as a diagnostic and prognostic marker of NPC in human patients. Next, they use cellular models of NPC to analyze the underlying mechanisms by which INSL5 regulates the proliferation and growth of these cancer cells. They show, using different approaches, that INSL5 through binding to its receptor GPCR142, activates the STAT5 pathway to regulate the expression of genes involved in the control of glycolysis. They claim that through this process, INSL5 facilitates the metabolic switch required for cancer cells to proliferate. The study is very extensive, and covers from studies in human patients of cancer to work in mouse models. Furthermore, through the use of cellular models they unravel the molecular mechanisms underlying the effects of INSL5 on cancer cells. This is a novel and original work that has been well conducted. There is no doubt about the relevance of this study for cancer research and treatment. Some additional work, as is next suggested, would further improve the quality of the

manuscript and reinforce the conclusions.

1. The authors claim that the function of INSL5 in the proliferation and growth of NPC cancer cells is to promote the required metabolic switch. While this is well proven in this study, the authors cannot conclude that this is the only, or the most important mechanisms that underlie the effects of INSL5 in cancer progression. Indeed, in the figure 6F, the growth of the tumors is only minimally changed when glycolysis is inhibited by 2-DG.

Response:

Thank you very much. We agree with you that metabolic switch is not the only mechanism to underlie the effects of INSL5 in cancer progression. As your helpful suggestion, we also found that INSL5 could enhance cell cycle progression and suppress cell apoptosis (Fig S6). 2-DG had minimal effect on tumor growth, but in our revised manuscript we found that 2-DG could reverse the chemoresistance induced by INSL5, which indicated that 2-DG could sensitize INSL5 expressed NPC to chemotherapy (Fig 6F and Fig S7C). Based on those finding, besides to metabolic switch, we corrected our conclusion that INSL5 induced metabolism reprogramming at least partly contributed the accelerated proliferation in the discussion

2. The STAT5 pathway has been involved in tumorigenesis in several types of cancer. Previous studies show that STAT5 can mediate oncogenic signals and regulate cell cycle progression, proliferation and promote cancer cell survival. Are there other STAT5-regulated pathways, in addition to glycolysis, changed in the models used?

Response:

Thanks for your great suggestion. Indeed, STAT5 was involved in many oncogenic signals as you mentioned. In the revised manuscript we added the new data about cell cycle, which suggested that INSL5 overexpression could promote cell cycle

progression. We also detected the key cyclins and p27, and found that INSL5 overexpression could enhance cyclin D and cyclin E expression, and decreased p27 level (Fig S6A-B). All of those suggested that INSL5 could also promote cell cycle progression in addition to glycolysis.

3. Are JAK kinases, the upstream activators of the STAT pathway also involved? Are JAK kinase inhibitors or STAT5 knock-down experiments abrogate the effects of INSL5?

Response:

We really appreciate your suggestion. We detected several possible upstream activators (Akt, ERK1/2 and JAK) of STAT pathway and found that INSL5 could increase the phosphorylation of Akt, ERK1/2 as reported, also JAK1 and STAT5. Meanwhile, we treated the cell with different kinase inhibitor to detect the phosphorylation of STAT5 and found that JAK1 (Ruxolitinib) and ERK1/2 (U0126) inhibitors, not Akt inhibitor (MK2206), could reverse STAT5 activation (Fig 5E). JAK1 inhibitor and STAT5 knock-down could decrease INSL5 induced cell proliferation and glucose uptake (Fig 5H-J). Taken together, those data suggested that INSL5 activated STAT5 through the JAK kinases in the upstream and JAK kinase inhibitor or STAT5 knock-down could abrogate the pro-tumor effect of INSL5

4. Several reports suggest that there is an important correlation between the JAK/STAT pathway and cell metabolism, notably the activation of hypoxia-inducible factors and the alteration of mitochondrial activity. In addition to the regulation of glycolysis genes, STAT5 has also been involved in mitochondrial function. A more detailed functional evaluation of mitochondria and glycolytic function of the cells with or without INSL5 (using for instance Seahorse analysis) would further prove the direct involvement of this protein in the control of metabolism in these cancer cells.

Response:

Thanks for your helpful suggestion. We performed Seahorse analysis in three different INSL5 overexpressing cell lines (CNE1, CNE2 and HK1) to detect extracellular acidification rate (ECAR) and oxygen consumption rate (OCR) using a XF96 Extracellular Flux analyzer. We demonstrated that INSL5 overexpression significantly increased glycolysis (Fig 4F-G and Fig S5G) and impaired oxidative phosphorylation (OXPHOS) (Fig S5H), which indicated that INSL5 overexpression could promote metabolism shift from OXPHOS to aerobic glycolysis.

5. In the Figure 6 A-B, the authors show the effects of INSL5 inhibition by using a monoclonal antibody. Cell death and apoptosis should be analyzed in these cells.

Response:

We performed the same treatment as Figure 6A-B, and then detected the cell death and apoptosis by flow cytometry after annexin V/propidium iodide (PI) staining. The results showed that the monoclonal antibody along had no effects on cell death and apoptosis (Fig S7A).

25th May 2020

Thank you for the submission of your revised manuscript to EMBO Molecular Medicine. We have now received the enclosed report from the two referees who were asked to re-assess it. As you will see the referees are now supportive and I am pleased to inform you that we will be able to accept your manuscript pending the following amendments.

***** Reviewer's comments *****

Referee #1 (Comments on Novelty/Model System for Author):

The revised version of the manuscript has satisfactorily addressed my previous concerns about mechanistic aspects of the study.

Referee #1 (Remarks for Author):

The revised version of the manuscript has satisfactorily addressed my previous concerns about mechanistic aspects of the study.

Referee #2 (Remarks for Author):

The authors have properly addressed my concerns and suggestions. The manuscript has improved the quality.

Corresponding Author Name: Yu-Ming Zheng,Zhi-Qiang Ling,Jian-Hua Sui,Mu-Sheng Zeng

Manuscript Number: EMM-2020-12050-V3